# Wasserstein Distributionally Robust Policy Evaluation and Learning for Contextual Bandits

**Yi Shen** *yi.shen478@duke.edu*
*Duke University*

**Pan Xu** *pan.xu@duke.edu*
*Duke University*

**Michael M. Zavlanos** *michael.zavlanos@duke.edu*
*Duke University*

**Reviewed on OpenReview:** *https://openreview.net/forum?id=NmpjDHWIvg*

## Abstract

Off-policy evaluation and learning are concerned with assessing a given policy and learning an optimal policy from offline data without direct interaction with the environment. Often, the environment in which the data are collected differs from the environment in which the learned policy is applied. To account for the effect of different environments during learning and execution, distributionally robust optimization (DRO) methods have been developed that compute worst-case bounds on the policy values assuming that the distribution of the new environment lies within an uncertainty set. Typically, this uncertainty set is defined based on the KL divergence around the empirical distribution computed from the logging dataset. However, the KL uncertainty set fails to encompass distributions with varying support and lacks awareness of the geometry of the distribution support. As a result, KL approaches fall short in addressing practical environment mismatches and lead to over-fitting to worst-case scenarios. To overcome these limitations, we propose a novel DRO approach that employs the Wasserstein distance instead. While Wasserstein DRO is generally computationally more expensive compared to KL DRO, we present a regularized method and a practical (biased) stochastic gradient descent method to optimize the policy efficiently. We also provide a theoretical analysis of the finite sample complexity and iteration complexity for our proposed method. We further validate our approach using a public dataset that was recorded in a randomized stoke trial.

## 1 Introduction

Contextual bandits (Langford & Zhang, 2007) are a class of decision-making problems where a learner repeatedly observes a context, takes an action from a set of candidate actions, and receives a cost for the chosen action. This cost signal is often referred to as bandit feedback since no costs for the unchosen actions are observed. The goal for the learner is to minimize the cost by selecting the best action for each context. Many problems in practice can be modeled as contextual bandits, ranging from online news recommendation (Li et al., 2011), advertising (Bottou et al., 2013) to personalized healthcare (Zhou et al., 2017).

In many high-stakes applications, such as healthcare, it is generally unsafe and not recommended to interact with the environment directly in order to collect data and evaluate or learn a desired policy. Instead, in these environments, observational data are often available, that have been collected by safe or risk-averse policies that depend on different contexts, e.g., policies that assign placebos to high-risk patients when testing new drugs. In this paper, we study off-policy evaluation (OPE) and off-policy learning (OPL) using observational data (Dudík et al., 2014). OPE estimates the average cost of following a given policy, while OPL aims to

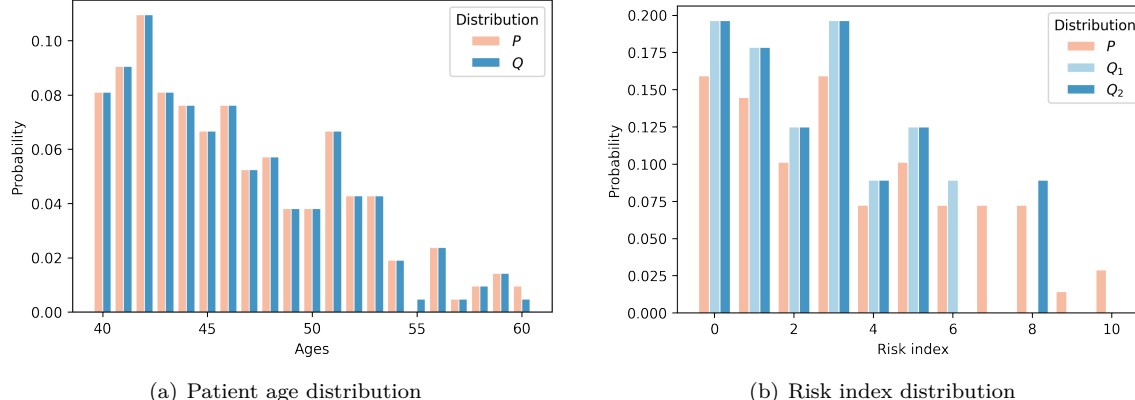

(a) Patient age distribution

(b) Risk index distribution

Figure 1: Two artificial datasets that represent patients' contextual distributions, where $x$ axis is the context support and $y$ axis is the probability. In (a), the distributions $P$ and $Q$ represent the patients' ages and are only different at the ages 55 and 60. The KL divergence is $\mathrm{KL}(Q||P) = +\infty$ since the two distributions have different supports, where the KL divergence between two discrete distributions $P$ and $Q$ is defined as $\mathrm{KL}(Q||P) = \sum_{x \in \mathcal{X}} Q(x)(\log(Q(x)/P(x))$. In (b), the distributions $Q_1$ and $Q_2$ represent the patients' risk index, the higher the worse. They are equally distant from $P$ under the KL divergence, i.e., $\mathrm{KL}(Q_1||P) = \mathrm{KL}(Q_2||P) = 0.21$. However, the distribution $Q_1$ represents a less challenging environment compared to the nominal distribution $P$ as the possibilities of encountering patients with higher risks in $Q_1$ are smaller than both $P$ and $Q_2$. $Q_2$ represents a similar environment as $P$ and is indeed closer to $P$ than $Q_1$ under the Wasserstein distance, i.e., $W(P, Q_1) = 2.07$ and $W(P, Q_2) = 1.42$. See equation 2.2 for the Wasserstein distance definition.

find the optimal policy that achieves the minimum cost. Generally, both OPE and OPL assume that the environment in which the observational are collected (training set) and the environment in which the trained policy is deployed (testing set) are identical. This means that the distributions of both contexts and costs do not change. However, in practice, this assumption is often violated. For instance, in clinical trials (Finlayson et al., 2021; Imai & Ratkovic, 2013), treatments that are tested and found effective using Randomized Clinical Trials (RCT) are not necessarily equally effective in local hospitals since the patient populations can be significantly different; RCTs use stringent inclusion and exclusion criteria to select the patient population while local hospitals reflect the local care environment, determined, e.g., by local social determinants of health. These differences between the training and testing environments are termed distribution shifts.

To address the effect of distribution shifts on OPE and OPL, distributionally robust optimization (DRO) methods have been recently developed (Si et al., 2020; Kallus et al., 2022; Mu et al., 2022), that aim to obtain worst-case bounds on the true OPE and OPL costs by assuming that the testing distributions lie within an uncertainty set around the training distributions. This way, over-optimism in unknown environments is avoided. Si et al. (2020) proposed importance sampling methods to solve both OPE and OPL problems, assuming that the behavior policy used to collect the observational data is known. They also provided an asymptotic convergence result, demonstrating that the proposed estimator converges to the true values at a rate of $\mathcal{O}_p(n^{-1/2})$, where $n$ is the number of data samples. Kallus et al. (2022) further proposed doubly robust methods that relax the assumption of knowing the behavior policy and achieve the same convergence rate, even when the unknown behavior policy is non-parametrically estimated. A common assumption in the previous work (Si et al., 2020; Kallus et al., 2022) is that potential distribution shifts appear across both context and cost distributions. Mu et al. (2022) argue that this assumption can lead to conservative performance, and the DRO objective may degenerate to a non-robust solution, especially when features are continuous and the loss function is binary (Hu et al., 2018). To address this issue, Mu et al. (2022) proposed a factored DRO formulation that treats distributional shifts in contexts and costs separately. This approach provides non-asymptotic sample complexity results.

All previous works consider the distribution shifts under Kullback-Leibler divergence (KL), i.e., the size of the uncertainty set is measured using the KL divergence. DRO problems based on the KL divergence enjoy

a simple dual formulation with a scalar dual variable and strong duality holds under mild conditions; see Hu & Hong (2013) for details. However, a KL uncertainty set requires a strong assumption on the testing distributions, namely, that they are absolutely continuous with respect to the training (nominal) distribution. This assumption excludes distributions with different supports compared to the training distribution and can be often violated in practice; see Figure 1(a) for an example. In addition, using the KL divergence it is not straightforward to obtain an initial estimate of the radius of the uncertainty set. One approach to determine an initial radius of the uncertainty set is to divide training set into two subsets and compute the distance between them. This approach is not possible using the KL divergence due to potential differences in the supports of these subsets, that will result to the radius being infinite. Finally, it is important to note that KL divergence cannot capture the geometric differences between distributions. For example, it is possible that two distinct testing distributions have equal KL distances from the training distribution, but one of them represents a more significant distribution shift compared to the other. A visual illustration of this scenario can be observed in Figure 1(b). From a computational perspective, when the training set is large, stochastic gradient descent (SGD) appears to be a promising method to solve KL DRO problems, especially OPL problems since as it allows to iteratively search for the best policy. Yet, as mentioned by Namkoong & Duchi (2016), SGD generally fails to converge because the variance of the objective and its subgradients explodes as the dual variable approaches 0. The above limitations motivate the study of DRO problems using different measures to define the uncertainty set.

Two popular measures, alternative to the KL divergence, that have been used to characterize the uncertainty sets in DRO literature are uncertainty sets based on moment constraints (Delage & Ye, 2010; Zymler et al., 2013; Yang et al., 2023) and uncertainty sets defined by the Wasserstein distance (Mohajerin Esfahani & Kuhn, 2018; Zhao & Guan, 2018; Gao & Kleywegt, 2022). Moment constraints generally require the testing distribution to have similar moments to the training distribution, such as means and variances (first and second moments). Finite order moment constraints can be extended to infinite orders and have been explored in the context of Kernel DRO (Zhu et al., 2021), where the kernel function defines a metric between distributions, known as the maximum mean discrepancy (MMD) (Gretton et al., 2012). However, it has been observed that in optimal transport problems (Feydy et al., 2019), the gradient vanishes when evaluating the distance between two distributions under the MMD distance, leading to slow convergence rates in practice. Detailed examples and discussions can be found in Feydy (2020, Chapter 3.2). In contrast, the Wasserstein distance has been shown to result in fast and stable convergence in many optimal transport and machine learning problems (Frogner et al., 2015; Arjovsky et al., 2017). The Wasserstein distance considers the underlying geometry of the distributions, including their supports, and therefore allows to compare testing and training distributions with different supports, something that KL methods are not able to do. While Wasserstein DRO has been used to quantify uncertainty in various learning tasks including classification and regression (Kuhn et al., 2019), its application to OPE and OPL problems is mostly unexplored yet with a few recent exceptions (Kido, 2022; Adjaho & Christensen, 2022). Though Kido (2022) measures the distribution shifts by Wasserstein distance, this work assumes that the context distribution of the testing set is absolutely continuous with respect to the context distribution of the training set, i.e, the same assumption as in methods that rely on KL (Si et al., 2020; Mu et al., 2022). This assumption simplifies the dual problem but excludes potential shifts of different supports. The approach in Adjaho & Christensen (2022) considers OPL problems with binary actions and allows for distribution shifts of different supports; simple and interpretable policies are studied due to the assumptions that actions are binary, e.g., a threshold policy. However, finite sample analysis is not the focus of this study.

In this paper, we focus on OPE and OPL for contextual bandit problems under distribution shifts. Specifically, we assume that shifts occur in both the context and the cost distributions for each context-action pair and employ Wasserstein DRO to develop worst-case bounds on policy values. For OPE problems, we first calculate upper bounds on cost values that account for cost shifts in each context-action pair. These bounds are then used to determine upper bounds on the policy values that account for context shifts. We provide a finite sample analysis of our proposed method and demonstrate that the policy value obtained from a dataset of size $n$ converges to the true value at a rate of $\mathcal{O}_p(n^{-1/2})$. Compared to OPE problems that rely on KL DRO (Si et al., 2020; Kallus et al., 2022; Mu et al., 2022), the use of Wasserstein DRO introduces an inner maximization sub-problem, that increases computational cost. To overcome this challenge, we propose a regularized DRO method that approximates the inner maximization sub-problem using smoothing. For

OPL problems, we demonstrate that both Wasserstein DRO and regularized Wasserstein DRO methods converge to the optimal policy at the rate of $\mathcal{O}_p(n^{-1/2})$. The regularized DRO method improves the query complexity of the Wasserstein DRO method in finding a $\delta$-accurate solution by an order of $\mathcal{O}(\delta^{-1})$ when the OPL problem is convex. Since the query complexity of both Wasserstein DRO and regularized Wasserstein DRO methods depends on the size of the support of the distributions, we also propose a (biased) SGD method that can find the optimal policy with query complexity independent of the distribution support size. It is worth noting that previous works (Si et al., 2020; Kallus et al., 2022; Mu et al., 2022) do not provide explicit query complexities. In our study, we validate the proposed methods using a randomized controlled trial dataset that study the effects of drug treatment on acute ischemic stroke, further demonstrating their effectiveness.

## 2    Problem definition and preliminaries

Denote $\mathcal{X} \subset \mathbb{R}^m$ the set of contexts and $\mathcal{A} = \{a_1, \cdots, a_k\}$ the set of discrete actions. Let $\mathcal{D}_n := \{(X_i, A_i, Y_i)\}_{i=1}^n$ be the observational dataset of size $n$. We assume the dataset is generated as follows. The contexts $\{X_i\}_{i=1}^n$ are independent and identically distributed (i.i.d.) samples from a fixed context distribution $P_x^0$. Actions $\{A_i\}_{i=1}^n$ are selected according to a context-dependent behavior policy $\pi_0$, i.e, $A_i$ is sampled from the distribution $\pi_0(\cdot|X_i)$. After taking the action $A_i$, a random cost $Y_i$ is announced and it is an i.i.d sample from a fixed cost distribution $P_{X_i, A_i}^0$.

**Assumption 2.1** *We make the following assumptions on the behavior policy $\pi_0$, the cost functions and the contexts.*

1. *Sufficient exploration: $\pi_0(A|X) \geq \mu_{\pi_0} > 0$ for all context action pairs;*

2. *Discrete and bounded costs: $\forall x \in \mathcal{X}, a \in \mathcal{A}$, $\Xi_{x,a}$ is discrete, finite, and known. Moreover, $0 \leq y_{x,a}(\xi) \leq Y_{\max}$, $\forall \xi \in \Xi_{x,a}$. For simplicity, we assume $\Xi_{x,a} = \Xi$ for all $(x, a) \in \mathcal{X} \times \mathcal{A}$;*

3. *Discrete contexts with positive mass: the context set $\mathcal{X}$ is a finite and known discrete set and the probability mass of each context is bounded from below, i.e., there exists an $\mu_{x_0} > 0$ such that $P(X = x) \geq \mu_{x_0}$ for all $x \in \mathcal{X}$.*

The first two assumptions in Assumption 2.1 are standard in non-robust OPE and OPL problems (Dudík et al., 2014; Wang et al., 2017) and are adopted in the KL DRO problems (Si et al., 2020; Kallus et al., 2022; Mu et al., 2022). The last assumption together with the first assumption in Assumption 2.1 ensure that the observational dataset contains sufficient samples across all context and action pairs, which is also assumed in Mu et al. (2022). We further make the following assumption on the random costs $Y$.

**Assumption 2.2** *For each context action $(x, a)$ pair, we are given a measurable function $y_{x,a}(\xi) : \Xi \to \mathbb{R}$ and $\Xi \subset \mathbb{R}^m$ is known. The observational dataset is provided in the form of $\mathcal{D}_n := \{(X_i, A_i, \xi_i, Y_i)\}_{i=1}^n$, where $\xi_i$ is an i.i.d sample from a fixed distribution $P_{\Xi_{X_i, A_i}}^0$ and the cost $Y_i = y_{X_i, A_i}(\xi_i)$ is then calculated from the given function $y_{x,a}$.*

Assumption 2.2 holds in many applications. For example, in clinical trials, $\xi$ can represent a patient's health condition after the treatment, e.g., blood pressure, which is random and can be monitored; a cost function is defined according to this health condition. Alternatively, in news recommendation systems, $\xi$ can represent a user's behavior after receiving a batch of news, e.g., clicking or not, time spent on the page. A cost function can be designed to encourage specific user behaviors, e.g., $y(\text{click}) = -1$ and $y(\text{not click}) = 1$.

If the random cost $Y$ is observed directly from the environment, one can set $y_{x,a}(\xi) = \xi$ which is an identity function mapping the random cost to itself or set $y_{x,a}(\xi) = Y_{\max}$ for all $\xi$ that are not observed in the observational dataset. We note, however, that our analysis here applies to general cost functions that only need to satisfy Assumption 2.2. We note also that Assumption 2.2 subsumes the special case when $y_{x,a}(\xi) = y(\xi)$ for all $(x, a)$ pairs and a given cost function $y(\xi)$, i.e., the case when the cost function only depends on the random observation $\xi$, as the examples discussed above. Thus, knowing the cost function does not imply knowing the relations between different $(x, a)$ pairs.

We assume distribution shifts happen both in the distribution of contexts and the distributions of the cost functions. Specifically, denote by $P_x$ the distribution of the contexts in the testing environment and assume that $P_x$ is contained in a uncertainty set $\mathcal{U}(\epsilon_x; P_x^0)$ that is centered around the training distribution $P_x^0$, i.e., $P_x \in \mathcal{U}(\epsilon_x; P_x^0) = \{P : D(P, P_x^0) \leq \epsilon_x\}$ where $\epsilon_x > 0$ and $D$ is a distance metric to be specified later. Similarly, for each context and action pair $(x, a)$, we assume the distribution of the cost in the testing environment satisfies $P_{x,a} \in \mathcal{U}(\epsilon_c; P_{x,a}^0)$ where $\epsilon_c > 0$.

Due to the possible distributional shifts in $P_x^0$ and $P_{x,a}^0$, we want to be conservative when evaluating or learning a new policy in the testing environment. To do so, we focus on the worst-case value of the new policy obtained over all possible distributions within the uncertainty sets.

**Definition 2.3** *For given $\epsilon_x, \epsilon_c > 0$ and policy $\pi$, define the distributionally robust policy value $V(\pi)$ as:*

$$V(\pi) = \sup_{P_x \in \mathcal{U}(\epsilon_x; P_x^0)} \mathbb{E}_{x \sim P_x} \left[ \mathbb{E}_{a \sim \pi(\cdot|x)} \left[ \sup_{P_{x,a} \in \mathcal{U}(\epsilon_c; P_{x,a}^0)} \mathbb{E}_{P_{x,a}} [Y] \right] \right]. \tag{2.1}$$

**Remark 2.4** *The formulation in equation 2.1 implies that given any two context and action pairs $(x, a), (x', a')$ and their training cost distributions $P_{x,a}^0, P_{x',a'}^0$, the testing cost distributions $P_{x,a}, P_{x',a'}$ are independent. Thus, the inner maximization problem is not coupled across contexts or actions and can be solved individually for each context and action pair.*

In this paper, we employ the Wasserstein distance as a distance metric between two probability distributions $P$ and $Q$, defined as

$$D(P, Q) := \inf_{\sigma} \left\{ \mathbb{E}_{(\xi,\zeta) \sim \sigma} [c(\xi, \zeta)] : \sigma_1 = P, \sigma_2 = Q \right\}, \tag{2.2}$$

where $c(\xi, \zeta)$ is a continuous function that captures the cost of moving the point $\xi$ to $\zeta$. In what follows, we assume $c(\xi, \zeta) = (\xi - \zeta)^2$. In equation 2.2, $\sigma$ is the coupling distribution with its first marginal $\sigma_1$ and second marginal $\sigma_2$ being $P$ and $Q$, respectively. The optimal coupling distribution $\sigma$ provides the best transportation plan that moves $P$ to $Q$ with respect to the cost function $c(\cdot, \cdot)$.

We now present some preliminary results on the DRO using the Wasserstein distance. Specifically, given a nominal distribution $P^0$ with support $\mathcal{X}$, a loss function $f(x)$, and a uncertainty parameter $\epsilon$, the (primal) Wasserstein DRO problem is defined as

$$(P) = \sup_{P \in \mathcal{U}(\epsilon; P^0)} \mathbb{E}_{x \sim P} [f(x)], \tag{2.3}$$

where $\mathcal{U}(\epsilon; P^0) = \{P \in \mathcal{M}^+ : \inf_{\sigma} \{\mathbb{E}_{(\xi,\zeta) \sim \sigma} [c(\xi, \zeta)] : \sigma_1 = P^0, \sigma_2 = P\} \leq \epsilon\}$ and $\mathcal{M}^+$ represents the set of all probability measures. Observe that solving the primal DRO problem is challenging since the optimization variable is a measure function and the expectation is taken with respect to an unobserved distribution $P$. This difficulty can be avoided if the DRO problem 2.3 is studied in its dual form, where the optimization variable becomes a scalar. In fact, Zhao & Guan (2018) showed that, under mild conditions, strong duality holds, i.e., zero duality gap; see Proposition 2 thereof for the details. The dual Wasserstein DRO problem can be formulated as

$$(D) = \inf_{\lambda \geq 0} \left\{ \epsilon \lambda + \mathbb{E}_{x \sim P^0} \left[ \sup_{\zeta \in \mathcal{X}} \left( f(\zeta) - \lambda(x - \zeta)^2 \right) \right] \right\}. \tag{2.4}$$

Indeed, in this dual problem, the optimization variable is a scalar and the expectation is taken with respect to the given nominal distribution $P^0$. Recent literature on policy evaluation has studied this dual DRO problem (Si et al., 2020; Mu et al., 2022), but using the KL divergence to characterize the uncertainty set (Hu & Hong, 2013). In this case, the dual KL DRO problem becomes:

$$(D_{\text{KL}}) = \inf_{\lambda \geq 0} \left\{ \epsilon \lambda + \lambda \ln \mathbb{E}_{x \sim P^0} [\exp(f(x)/\lambda)] \right\}. \tag{2.5}$$

As discussed before, compared to the Wasserstein DRO problem $(D)$, the KL DRO problem, $(D_{\text{KL}})$ tends to up-weight the samples with high costs and may over-fit to outliers since it is not aware of the geometry of the

distribution. See Appendix E for an example that compares KL DRO and Wasserstein DRO. Nevertheless, the Wasserstein dual problem ($D$) is computationally more challenging as it involves an inner maximization problem.

The following lemma shows that if $f(\cdot)$ is bounded, the optimal solution to the dual problem ($D$) is finite. This avoids limit arguments in the proofs that involve the dual variable $\lambda$.

**Lemma 2.5** *Consider the dual problem in equation 2.4. If $0 \leq f(x) \leq f_{\max}$ for all $x \in \mathcal{X}$, then the optimal solution to equation 2.4 satisfies $\lambda^* \in [0, f_{\max}/\epsilon]$ and the optimal value of equation 2.4 attained at $\lambda^*$ satisfies $D^* \leq f_{\max}$.*

The proof is provided in Appendix B.

## 3 Wasserstein DRO for policy evaluation

In this section, we focus on the OPE problem equation 2.1 with the uncertainty set described using the Wasserstein distance metric.

**Proposition 3.1** *Let Assumption 2.2 hold. The policy evaluation problem equation 2.1 is equivalent to the following two-step dual problem.*

$$V(\pi) = \inf_{\lambda \geq 0} \left\{ \epsilon_x \lambda + \mathbb{E}_{x \sim P_x^0} \left[ \sup_{\zeta \in \mathcal{X}} \left( \mathbb{E}_{a \sim \pi(\cdot|\zeta)} \left[ m(\zeta, a) \right] - \lambda (x - \zeta)^2 \right) \right] \right\}, \tag{3.1}$$

$$\text{where } m(x, a) = \inf_{\lambda \geq 0} \left\{ \epsilon_c \lambda + \mathbb{E}_{\xi \sim P_{\Xi_{x,a}}^0} \sup_{\zeta \in \Xi_{x,a}} \left( y_{x,a}(\zeta) - \lambda (\xi - \zeta)^2 \right) \right\}, \quad \forall x \in \mathcal{X}, a \in \mathcal{A}. \tag{3.2}$$

**Proof** The result follows by applying the duality as in equation 2.4 to both contexts and costs. ∎

Recall that Assumption 2.2 poses an extra condition on the cost functions. From the dual problem perspective, we observe that this assumption is necessary for addressing distribution shifts in Wasserstein DRO as the dual problem equation 3.2 contains an inner maximization problem, which requires the knowledge of the function values for all realizations. Additional assumptions on the cost functions are required when applying KL DRO to OPE and OPL problems, where the probability mass function $P(Y = y|x, a)$ is lower bounded for all $y$ across all context and action pairs (Si et al., 2020; Kallus et al., 2022; Mu et al., 2022). Similarly, this lower bound assumption ensures that the dual problem under KL is well defined.

Note that, in general, it is intractable to directly solve equation 3.1 and equation 3.2 since these problems involve an inner maximization problem that is hard to solve. The inner maximization problem can be solved efficiently assuming, e.g., that $\mathcal{X}$ and $\Xi$ are discrete. Besides, the distributions $P_x^0$ and $P_\Xi^0$ in equation 3.1 and equation 3.2 are unknown and can only be estimated from the finite sample dataset $\mathcal{D}_n$.

Given the observational dataset $\mathcal{D}_n$, denote $\hat{P}_x^0$ and $\{\hat{P}_{\Xi_{x,a}}^0\}_{x \in \mathcal{X}, a \in \mathcal{A}}$ the empirical distribution estimates of $P_x^0$ and $\{P_{\Xi_{x,a}}^0\}_{x \in \mathcal{X}, a \in \mathcal{A}}$, respectively. We define the empirical policy evaluation problem as follows:

$$\hat{V}(\pi) = \inf_{\lambda \geq 0} \left\{ \epsilon_x \lambda + \mathbb{E}_{x \sim \hat{P}_x^0} \left[ \sup_{\zeta \in \mathcal{X}} \left( \mathbb{E}_{a \sim \pi(\cdot|\zeta)} \left[ \hat{m}(\zeta, a) \right] - \lambda (x - \zeta)^2 \right) \right] \right\}, \tag{3.3}$$

$$\text{where } \hat{m}(x, a) = \inf_{\lambda \geq 0} \left\{ \epsilon \lambda + \mathbb{E}_{\xi \sim \hat{P}_{\Xi_{x,a}}^0} \left[ \sup_{\zeta \in \Xi_{x,a}} \left( y_{x,a}(\zeta) - \lambda (\xi - \zeta)^2 \right) \right] \right\}, \, \forall x \in \mathcal{X}, a \in \mathcal{A}. \tag{3.4}$$

The goal of the OPE problem is to estimate the policy value $V(\pi)$. As we can only estimate the empirical policy value $\hat{V}(\pi)$ using the offline data in practice, in the following theorem, we provide a finite sample analysis of the gap between $\hat{V}(\pi)$ and $V(\pi)$.

**Theorem 3.2** *Given an evaluation policy $\pi$, let Assumptions 2.1 and 2.2 hold and select the total number of samples as $n \geq 2\log(4|\mathcal{X}||\mathcal{A}|/\delta)/(\mu_{\pi_0}\mu_{x_0})^2$. Then, with probability $1 - \delta$, the following inequality holds,*

$$
\begin{aligned}
|\hat{V}(\pi) - V(\pi)| &\leq Y_{\max}\left( \sqrt{\frac{2\log(2/\delta)}{n}} + \sqrt{\frac{|\mathcal{X}|}{n}} + \sqrt{\frac{4\log(4|\mathcal{X}||\mathcal{A}|/\delta)}{n\mu_{\pi_0}\mu_{x_0}}} + \sqrt{\frac{2|\Xi|}{n\mu_{\pi_0}\mu_{x_0}}} \right) \\
&= \mathcal{O}\left( \sqrt{\frac{\log(|\mathcal{X}||\mathcal{A}|/\delta)}{n}} \right) + \mathcal{O}\left( \sqrt{\frac{|\mathcal{X}| + |\Xi|}{n}} \right).
\end{aligned}
\tag{3.5}
$$

The proof is provided in Appendix A.1. Theorem 3.2 shows that the policy evaluation error decreases at a rate of $n^{-1/2}$. Note that Mu et al. (2022) consider a similar problem using the KL divergence to characterize the uncertainty set. In this case, they show that the policy evaluation error satisfies $|\hat{V}(\pi) - V(\pi)| = \mathcal{O}(\sqrt{(\log(n)\log(|\mathcal{X}||\mathcal{A}|/\delta))/(\epsilon_c n)})$, where $\epsilon_c$ is the KL uncertainty set radius.

Our error bound in Theorem 3.2 contains an extra term $\mathcal{O}(\sqrt{(|\mathcal{X}| + |\Xi|)/n})$ that is due to the use of Wasserstein distance and is directly related to the total variation of the empirical discrete distribution compared to the true discrete distribution. This error increases if the size of the supports of $X$ and $\Xi$ increases as the Wasserstein distance is a geometry-aware metric. Comparing the first term in equation 3.5 to the KL error bounds provided in Mu et al. (2022), we observe that our result improves the error bound by an order of $\sqrt{\log(n)}$. We believe that the extra $\sqrt{\log(n)}$ term in the error bound in Mu et al. (2022) is caused by the corresponding analysis rather than the different characterizations of the uncertainty set; the KL error bound can be improved by using a similar analysis as in Lemma 5 in Zhou et al. (2021), which studies a distributionally robust reinforcement learning problem using a KL uncertainty set.

Note that the KL error bound additionally depends on the radius of the uncertainty set $\epsilon_c$, which is due to the fact that the dual variable $\lambda$ multiplies the log-expectation term as in equation 2.5. This KL error bound increases when the radius of the uncertainty set decreases. Although this error bound is always upper bounded by $Y_{\max}$, it becomes uninformative when the distributional shift is small. This is because $\mathcal{O}(\sqrt{(\log(n)\log(|\mathcal{X}||\mathcal{A}|/\delta))/(\epsilon_c n)}) \gg Y_{\max}$ when $\epsilon_c \to 0$. On the other hand, this KL error bound decreases as the radius of the uncertainty set increases. This is because when $\epsilon_c$ is large, the optimal dual variable $\lambda$ in KL dual problem is small and thus the value of $\lambda \ln \mathbb{E}_{x \sim P^0}[\exp(f(x)/\lambda)]$ converges to the highest cost. In contrast, our proposed Wasserstein error bound in equation 3.5 does not depend on the uncertainty radius.

Concurrently with the method proposed here, Xu et al. (2023) studied a DRO problem for a robust reinforcement learning problem also using the Wasserstein distance. However, in Xu et al. (2023) the cost function is known and deterministic while here we consider stochastic costs with distributional shifts. Besides, the error bound provided in Xu et al. (2023) is radius-dependent and will not be informative when the radius is small, similar to the KL formulation in Mu et al. (2022) discussed above. We believe that this dependency is due to the covering number type of arguments in the analysis, which is designed for improving state dependency in the sample complexity results. A reinforcement learning OPE problem using Wasserstein DRO has recently been studied in Wang et al. (2021), assuming that the behavior policy that collects the training set is known or can be estimated accurately. This known behavior policy is directly used in the primal and dual formulations. More importantly, to achieve zero duality gap, extra assumptions on this behavior policy are required as stated in their Theorem 1 in Wang et al. (2021). These extra assumptions cannot be verified for the problem considered here that only assumes knowledge of the finite training set.

In practice, $\epsilon_c$ can be selected proportional to the sample size to capture uncertainties caused by finite samples (Ramdas et al., 2017), e.g., $\epsilon_c = \mathcal{O}(n^{-1/2})$. Then, the policy evaluation error decreases at the rate of $n^{-1/4}$ under KL divergence while the rate under Wasserstein distance in equation 3.5 remains the same of $n^{-1/2}$.

## 4 Regularized Wasserstein DRO for policy evaluation

The empirical policy evaluation problem under the Wasserstein distance can be reformulated as a linear program (LP) and thus can be solved using commercial linear programming solvers, e.g., Gurobi (Gurobi Optimization, LLC, 2023). However, as often discussed in much of the optimal transport literature (Cuturi,

2013; Genevay et al., 2016; Weed, 2018), calculating the Wasserstein distance between two distributions is computationally expensive when the support set is large, e.g., when $|\mathcal{X}|$ or $|\Xi|$ is large. In this case, entropy-regularized methods have been proposed to reduce the computational complexity of optimal transport problems. Inspired by these methods, we simplify the dual problem in equation 2.4 by replacing the maximization operation with a smooth "softmax" function, e.g., the log-sum-exp function. Specifically, we define the smoothed dual problem as:

$$(D_\eta) := \inf_{\lambda \geq 0} \left\{ \epsilon\lambda + \mathbb{E}_{x \sim P_x^0}\left[ 1/\eta \log \left( \sum_{\zeta \in \mathcal{X}} 1/|\mathcal{X}| e^{\eta(f(\zeta) - \lambda(x-\zeta)^2)} \right) \right] \right\}, \tag{4.1}$$

where $\eta > 0$ is a hyper-parameter that controls the distance between the "softmax" and the maximum. The primal problem of the smoothed dual problem $(D_\eta)$ is indeed a DRO problem but with an entropy-regularized objective function. See Appendix D for the duality connection.

**Remark 4.1** *Although the smoothed dual problem $(D_\eta)$ and dual problem $D_{KL}$ in equation 2.5 may look similar to each other, they are in fact fundamentally different. Specifically, in $D_{KL}$, the dual variable $\lambda$ directly affects the exponential function inside the expectation, e.g., a small $\lambda$ up-weights regions of $\mathcal{X}$ with high costs $f(x)$. On the other hand, in $(D_\eta)$, $\eta$ is a pre-specified tuning parameter and the dual variable $\lambda$ is directly related to the geometry of the domain $\mathcal{X}$, e.g., for a given $x$, a large $\lambda$ only up-weights regions of $\mathcal{X}$ with high costs that are close to $x$. In addition, $(D_\eta)$ simplifies the theoretical analysis since the "softmax" function in $(D_\eta)$ is not be unbounded when $\lambda$ goes to $0$. To avoid this technical challenge, Mu et al. (2022) instead assume that the $\lambda$ in $D_{KL}$ is bounded from below.*

We define the empirical policy evaluation problem using the regularized dual problem as follows:

$$\hat{V}_\eta(\pi) = \inf_{\lambda \geq 0} \left\{ \epsilon_x \lambda + \mathbb{E}_{x \sim \hat{P}_x^0}\left[ 1/\eta \log \left( \sum_{\zeta \in \mathcal{X}} 1/|\mathcal{X}| e^{\eta(\mathbb{E}_{a \sim \pi(\cdot|\zeta)}[\hat{m}_\eta(\zeta,a)] - \lambda(x-\zeta)^2)} \right) \right] \right\}, \tag{4.2}$$

$$\text{where } \hat{m}_\eta(x,a) = \inf_{\lambda \geq 0} \left\{ \epsilon_c \lambda + \mathbb{E}_{\xi \sim \hat{P}^0_{\Xi_{x,a}}}\left[ 1/\eta \log \left( \sum_{\zeta \in \Xi_{x,a}} 1/|\Xi_{x,a}| e^{\eta(y_{x,a}(\zeta) - \lambda(\xi-\zeta)^2)} \right) \right] \right\}. \tag{4.3}$$

In the following theorem, we provide a finite sample analysis on the gap between $\hat{V}_\eta$ and $V$ for the empirical policy evaluation problem equation 4.2 and equation 4.3 that uses the regularized dual method.

**Theorem 4.2** *Given an evaluation policy $\pi$, let Assumptions 2.1 and 2.2 hold and select the total number of samples as $n \geq 2\log(4|\mathcal{X}||\mathcal{A}|/\delta)/(\mu_{\pi_0}\mu_{x_0})^2$. Then, with probability $1 - \delta$, the following inequality holds,*

$$|\hat{V}_\eta(\pi) - V(\pi)| \leq Y_{\max}\left( \sqrt{2\log(2/\delta)/n} + \sqrt{|\mathcal{X}|/n} + \sqrt{4\log(4|\mathcal{X}||\mathcal{A}|/\delta)/(n\mu_{\pi_0}\mu_{x_0})} + \sqrt{2|\Xi|/(n\mu_{\pi_0}\mu_{x_0})} \right)$$
$$+ \log(|\mathcal{X}|)/\eta.$$

Theorem 4.2 implies that the entropy regularized OPE problem can find the policy value $\hat{V}_\eta(\pi)$ almost as efficiently as the un-regularized OPE problem in equation 3.3 and equation 3.4. The error term $\log(|\mathcal{X}|)/\eta$ is caused by the approximation of inner maximization by the "softmax".

## 5 Wasserstein DRO for policy learning

In this section, we consider the OPL problem using the Wasserstein distance to characterize the uncertainty. Let $\Pi$ be the policy space. Define the optimal distributionally robust policy $\pi^* = \arg\min_{\pi \in \Pi} V(\pi)$ with optimal value $V^* = V(\pi^*)$. Similarly, define the empirical optimal distributionally robust policy $\hat{\pi}^* = \arg\min_{\pi \in \Pi} \hat{V}(\pi)$ with optimal value $\hat{V}^* = \hat{V}(\hat{\pi}^*)$. In the following theorem, we provide a finite sample analysis of the gap between $V^*$ and $\hat{V}^*$.

**Theorem 5.1** *Given a learning policy space $\Pi$, let Assumptions 2.1 and 2.2 hold and select the total number of samples $n \geq 2\log(4|\mathcal{X}||\mathcal{A}|/\delta)/(\mu_{\pi_0}\mu_{x_0})^2$. Then, with probability $1 - \delta$, the following inequality holds,*

$$|\hat{V}^* - V^*| \leq Y_{\max}\left( \sqrt{2\log(2/\delta)/n} + \sqrt{|\mathcal{X}|/n} + \sqrt{4\log(4|\mathcal{X}||\mathcal{A}|/\delta)/(n\mu_{\pi_0}\mu_{x_0})} \right.$$
$$\left. + \sqrt{2|\Xi|/(n\mu_{\pi_0}\mu_{x_0})} \right).$$

The proof is provided in Appendix C.1. Note that the results of Theorem 5.1 can be easily extended to the entropy-regularized problem described in Section 3.

Theorem 5.1 assumes that we can solve the optimization problem exactly without specifying how. In practice, if we parameterize the policy $\pi$ by $\theta \in \Theta$, we can use first-order methods to find the optimal policy. Specifically, denote $\hat{l}(\theta, \zeta) = \mathbb{E}_{a \sim \pi_\theta(\cdot|\zeta)}\big[\hat{m}(\zeta, a)\big]$. Then, we can define the OPL problem for the Wasserstein and regularized Wasserstein DRO formulations as follows:

$$\hat{V}(\pi^*) = \inf_{\theta, \lambda \geq 0} \left\{ \mathbb{E}_{x \sim \hat{P}_x^0}\left[ \sup_{\zeta \in \mathcal{X}} \left( \hat{l}(\theta, \zeta) - \lambda(x - \zeta)^2 + \epsilon_x \lambda \right) \right] \right\}, \tag{5.1}$$

$$\hat{V}_\eta(\pi^*) = \inf_{\theta, \lambda \geq 0} \left\{ \mathbb{E}_{x \sim \hat{P}_x^0}\left[ \frac{1}{\eta} \log \left( \sum_{\zeta \in \mathcal{X}} \frac{1}{|\mathcal{X}|} e^{\eta\left(\hat{l}(\theta, \zeta) - \lambda(x - \zeta)^2\right)} \right) \right] + \epsilon_x \lambda \right\}. \tag{5.2}$$

In general, the regularized problem in equation 5.2 is preferred to equation 5.1 in solving for the optimal solution. For example, consider a specific where $\hat{P}_x^0$ is a point mass at $x_0$. Then, equation 5.1 is equivalent to $\inf_{(\theta, \lambda)} L(\theta, \lambda)$, where $L(\theta, \zeta) = \max_{\zeta \in |\mathcal{X}|} \{ f_\zeta(\theta, \lambda) \}$ and $f_\zeta(\theta, \lambda) = \hat{l}(\theta, \zeta) - \lambda(x_0 - \zeta)^2 + \epsilon_x \lambda$. Therefore, in this case, $L(\theta, \lambda)$ is the point-wise maximum of the set of functions $f_\zeta$ indexed by $\zeta$. Note that $L(\theta, \lambda)$ is not guaranteed to be smooth even if all functions $\{f_\zeta\}_{\zeta \in |\mathcal{X}|}$ are smooth. Thus, if we further assume that each $f_\zeta(\theta, \lambda)$ is convex in $(\theta, \lambda)$ and solve the problem using first-order methods, e.g., subgradient methods (Nesterov et al., 2018), it will require $\mathcal{O}(\delta^{-2})$ iterations to find a $\delta$-accurate optimal solution and each iteration requires $\mathcal{O}(|\mathcal{X}|)$ queries to evaluate $f_\zeta$ and $\nabla f_\zeta$. However, as discussed in Nesterov (2005), if the "softmax" is used to approximate the inner maximization problem, the smoothed problem $L_\eta(\theta, \lambda)$ is smooth when all functions $\{f_\zeta\}_{\zeta \in |\mathcal{X}|}$ are smooth, e.g., when $\nabla f_\zeta$ is $\mathcal{O}(\delta^{-1})$-Lipschitz continuous, and it only takes $\tilde{O}(\delta^{-1})$ iterations to find a $\delta$-accurate optimal solution to the original problem $L(\theta, \zeta)$ by selecting $\eta = \mathcal{O}(\log |\mathcal{X}|/\delta)$. For each iteration, it requires the same $\mathcal{O}(|\mathcal{X}|)$ of queries to calculate the gradient. Since querying the functions $f_\zeta$ is computational costly in practice, solving equation 5.2 reduces the query complexity from $\mathcal{O}(|\mathcal{X}|\delta^{-2})$ to $\tilde{O}(|\mathcal{X}|\delta^{-1})$ when $f_\zeta$ are convex and smooth and $\hat{P}_x^0$ is a point mass.

Note that although the regularized OPL problem equation 5.2 improves the query complexity by an order of $\mathcal{O}(\delta^{-1})$, the linear dependency on $|\mathcal{X}|$ remains. If $|\mathcal{X}|$ is large, e.g., if $|\mathcal{X}| \gg \delta^{-1}$, the computational advantage of solving the regularized DRO problem equation 5.2 compared to the un-regularized DRO problem equation 5.1 is marginal. To further improve the query complexity in $|\mathcal{X}|$, we propose a biased stochastic gradient descent (SGD) method to approximate the gradient of equation 5.2. Specifically, at each time step, we sample one point from $\hat{P}_x^0$ and sample multiple points uniformly from the set $\mathcal{X}$. We then construct a biased gradient estimate using those samples. Note that unlike the SGD methods typically used in the machine learning literature, the finite sample stochastic gradient estimate is biased due to the nonlinear logarithm function in equation 5.2. The full algorithm is presented in Algorithm 1.

**Theorem 5.2** *(Iteration and query complexity of Algorithm 1). Assume that the policy space $\Theta$ is compact and there exists a finite number $B > 0$ such that for any $x, y \in \mathcal{X}$, $\|x - y\|_2 \leq B$. i) Suppose that $\hat{l}(\theta, \zeta) = \sum_{i=1}^{k} \pi_\theta(a_i|\zeta)\hat{m}(\zeta, a_i)$ is convex in $\theta$ and $L_{\hat{l}}$-smooth in $\theta$ for any $\zeta \in \mathcal{X}$. Then, the number of iterations required to find a $\delta$-accurate solution is in the order $\mathcal{O}(\delta^{-2})$ iterations to find a $\delta$-accurate optimal solution and the total query complexity is $\mathcal{O}(\delta^{-3})$. ii) Suppose that $\hat{l}(\theta, \zeta)$ is $L_{\hat{l}}$-smooth in $\theta$ for any $\zeta \in \mathcal{X}$. Then, the number of iterations required to find a $\delta$-accurate stationary point is in the order $\mathcal{O}(\delta^{-4})$ iterations and the total query complexity is $\mathcal{O}(\delta^{-6})$.*

Theorem 5.2 provides query complexity of Algorithm 1 for both convex and nonconvex smooth functions. The regularized DRO problem is a special case of so-called conditional stochastic optimization (CSO) problems (Hu et al., 2020). The proof of Theorem 5.2 is directly adapted from Theorem 3.2 and 3.3 in Hu et al. (2020) and is provided in Appendix C.2. As finding the optimal query complexity is not the focus of this paper, we defer readers to Carmon et al. (2021); Hu et al. (2020); Wang et al. (2021) for related discussions.

---

**Algorithm 1** Policy learning using biased stochastic gradient descent

---

**Require:** Robust cost estimates $\hat{m}(s, a)$ for each $(s, a)$ pair; context uncertainty radius $\epsilon_x$; parameterized policy $\pi_\theta$, where $\theta \in \Theta$; number of iterations $T$; inner batch size $m_t$ and step size $\gamma_t$,; initial policy parameter $\theta_0$ and dual variable $\lambda_0$; smoothing parameter $\eta$; context support set $\mathcal{X}$ and nominal context distribution $\hat{P}_x^0$.

1: **for** $t = 0, \ldots, T - 1$ **do**
2:     Sample a single context $x_t$ from $\hat{P}_x^0$ and $m_t$ points $\{\zeta_j\}_{j=1}^{m_t}$ uniformly from the support set $\mathcal{X}$.
3:     Evaluate the function values $\{g(\zeta_j; \theta_t, \lambda_t)\}_{j=1}^{m_t}$, where $g(\zeta_j; \theta_t, \lambda_t) = e^{\eta\left(\hat{l}(\theta_t, \zeta_j) - \lambda_t(x_t - \zeta_j)^2\right)}$
4:     Calculate the biased stochastic gradients $\nabla_{\theta_t}\hat{V}_\eta(\theta_t, \lambda_t)$ and $\partial\hat{V}_\eta(\theta_t, \lambda_t)/\partial\lambda_t$ according to equation 5.3 equation 5.4:

$$\nabla_{\theta_t}\hat{V}_\eta(\theta_t, \lambda_t) = \frac{\sum_{j=1}^{m_t} g(\zeta_j; \theta_t, \lambda_t)\nabla_{\theta_t}\hat{l}(\theta_t, \zeta_j)}{\sum_{j=1}^{m_t} g(\zeta_j; \theta_t, \lambda_t)}, \tag{5.3}$$

$$\frac{\partial\hat{V}_\eta(\theta_t, \lambda_t)}{\partial\lambda_t} = -\frac{\sum_{j=1}^{m_t} g(\zeta_j; \theta_t, \lambda_t)(x_t - \zeta_j)^2}{\sum_{j=1}^{m_t} g(\zeta_j; \theta_t, \lambda_t)} + \epsilon_x. \tag{5.4}$$

5:     Update the policy parameter and the dual variable:

$$\theta_{t+1} = \Pi_\Theta\left(\theta_t - \eta_t\nabla_{\theta_t}\hat{V}_\eta(\theta_t, \lambda_t)\right), \quad \lambda_{t+1} = \max\left\{\lambda_t - \eta_t\partial\hat{V}_\eta(\theta_t, \lambda_t)/\partial\lambda_t, 0\right\}.$$

6: **end for**

---

## 6 Policy evaluation and learning for the international stroke trial

In this section we validate our regularized Wasserstein DRO methods for both OPE and OPL problems on the International Stroke Trial (IST) (Group et al., 1997) dataset.

IST is a randomized controlled trial that aims to study the effects of early administration of aspirin and/or heparin on the clinical course of acute ischaemic stroke. The IST dataset (Sandercock et al., 2011) includes 19,435 patients with logged contextual information such as age, gender, and level of consciousness before treatment admissions, as well as follow-up results on day 14, including the occurrence of recurrent stroke, pulmonary embolism, and death. We define two actions from the IST dataset: the treatment action of prescribing both aspirin and heparin (high or medium doses) ($a_1$) and the control action of of not administering any treatment, neither aspirin or heparin ($a_2$). The behavioral policy $\pi_0(a = a_1) = 0.5$. The cost function is calculated based on the recorded follow-up events on day 14.

To introduce distribution shifts, we split the dataset into a training set and a testing set, and we introduce a selection bias into the training set. Specifically, we randomly remove 50% of the patients in the training set who are not fully conscious. This creates a difference in the context distribution between the training set and the testing set, with the patients in the testing set being more likely to be unconscious before treatment than those in the training set. In this simulation, we only consider context shifts and assume that the cost functions do not change between the training and testing sets. Due to the limited number of data points in the IST dataset, we further use decision trees and binning methods to reduce the number of distinct contexts. The detailed experimental setup are provided in Appendix F.

We first consider the OPE problem for a random policy using the training set. Although any policy can be evaluated using the proposed methods, we select the random policy since it is the same as the behavioral policy, and thus the underlying truth can be estimated more accurately from the testing set, i.e., the importance ratio is always 1 (Dudík et al., 2014). One main challenge in DRO is the selection of the radius, as the final robust value depends on this choice. When similar datasets that include both the testing and training sets are available, the uncertainty set radius can be selected by calculating the distance between the testing and the training sets; however, this distance is not always well-defined under KL divergence. Specifically, for a large number of contexts, it is unlikely that all contexts have been observed for any two given finite datasets. Therefore, the two datasets are likely to have different supports, resulting in an unbounded KL radius. Since the Wasserstein distance is well-defined regardless of possible differences in the supports of the context distributions in the two datasets, we can split the training set into two parts and calculate the

Table 1: Policy evaluation and learning results

| Method / Uncertainty set radius ($\epsilon_{\mathrm{W}}$) | 0.03 | 0.05 | 0.1 |
|---|---|---|---|
| OPE: Factor KL DRO ($\epsilon_{\mathrm{KL}}$) | 0.30 (0.01) | 0.374 (0.1) | 0.74 (1.0) |
| OPE: Wasserstein DRO (LP on sub-support) | 0.53 | 0.61 | 0.79 |
| OPE: Regulated Wasserstein DRO (BSGD) | 0.59 | 0.76 | 1.05 |
| OPL: Factor KL DRO ($\epsilon_{\mathrm{KL}}$) | 0.28 (0.01) | 0.368 (0.1) | 0.69 (1.0) |
| OPL: Regulated Wasserstein DRO (BSGD) | 0.54 | 0.62 | 0.92 |
| Expectation under $\hat{P}$ (training set, random policy) | 0.28 | | |
| Expectation under $Q$ (testing set, random policy) | 0.38 | | |

Wasserstein distance between them, which yields the uncertainty set $\epsilon_{\mathrm{w}} = 0.03$ in this simulation. This distance captures the data uncertainty due to finite samples and provides a lower bound on the uncertainty set radius. To further capture distribution shifts, we should select the radius larger than this distance.

The policy evaluation results are presented in Table 1. Solving the OPE problem using the Wasserstein DRO method is challenging since the support size is large. Therefore, we approximate the solution by only considering observed contexts in training set instead of the full set $\mathcal{X}$. However, this Wasserstein DRO using the sub-support only provides a lower bound to the full support DRO method, as the inner maximum problem is approximated by maximizing over a smaller subset for a given radius. On the other hand, regularized Wasserstein DRO does not face the issue of a large support set, as it directly approximates the solution through sampling. It is important to observe that both Wasserstein DRO methods provide upper bounds to the true expectation under the testing distribution $Q$ as in Table 1. Since we consider separate uncertainty sets on both the contexts and the rewards, we adopt the Factor KL DRO method developed in Mu et al. (2022) to benchmark the performance of Wasserstein DRO, where similar separate uncertainty sets are considered. As no specific numerical algorithm is provided in Mu et al. (2022) to solve the KL DRO problem, we simply use a gradient descent method to find the optimal solution. Note that the DRO problem under KL divergence is convex, gradient descent is guaranteed to converge to the global optimal point. The gradient derivations of KL DRO can be found in the appendix of Mu et al. (2022). As estimating the uncertainty set radius for KL divergence is not straightforward, we test different radii when using KL DRO. However, it is worth noting that the KL DRO method fails to provide valid upper bounds when $\epsilon_{\mathrm{KL}} = 0.01$ or 0.1.

Finally, we consider the OPL problem by parameterizing a context-dependent policy. Specifically, if a patient is not fully conscious, there is a probability $\theta_1$ of taking action $a_1$, and if a patient is fully conscious, there is a probability $\theta_2$ of taking action $a_1$. The optimal policy values are shown in Table 1, where the Factor KL DRO is solved by gradient descent on both the policy parameters and the KL dual variable. Note that the optimal policies achieve lower costs compared to the random policy for both methods under all three uncertainty sets, indicating that the random policy is not robustly optimal. The optimal policy parameters of the regulated Wasserstein DRO converge to $\theta_1 = 0.8$ and $\theta_2 = 1$ when $\epsilon_{\mathrm{W}} = 0.03$, which suggest that the early administration of aspirin and/or heparin is effective on the clinical course of acute ischaemic stoke at the population level. Note that the optimal robust policy is not necessary deterministic as the value function in equation 5.1 is not linear in the policy parameters. Further results on policy learning can be found in Appendix F.

## 7    Conclusion

In this work, we proposed a distributionally robust optimization (DRO) framework using Wasserstein distance (Wasserstein DRO) to address distribution shifts in OPE and OPL problems. We provided a finite sample analysis and showed that the policy value obtained from a dataset of size $n$ converges to the true values at a rate of $\mathcal{O}_p(n^{-1/2})$ under the proposed framework. As Wasserstein DRO is computationally challenging when the support set of the underlying distribution is large, we further formulated a regularized Wasserstein DRO problem and provided a corresponding biased stochastic gradient descent algorithm that is

independent of the distribution support size and can numerically solve the regularized problem; the proposed algorithm achieves better iteration and query complexity compared to solving Wasserstein DRO using linear programs while maintaining the same convergence rate. We validated our proposed methods using a medical stroke trial dataset.

**Acknowledgments**

This work is supported in part by AFOSR under award #FA9550-19-1-0169 and by NSF under award CNS-1932011. Pan Xu is supported by the Whitehead Scholars Program and the Department of Biostatistics and Bioinformatics at Duke University.

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

# Appendix

# A    Proofs for Off-Policy Evaluation

## A.1    Proof of Theorem 3.2

In what follows, we first provide an error decomposition lemma.

**Lemma A.1** *Let Assumptions 2.2 and 2.1 hold. Denote $p(x_i) = P_x^0(X = x_i)$ and $\hat{p}(x_i) = \hat{P}_x^0(X = x_i)$. For any given policy $\pi$, define $l(\zeta) = \mathbb{E}_{a \sim \pi(\cdot | \zeta)} [m(\zeta, a)]$ and $\hat{l}(\zeta) = \mathbb{E}_{a \sim \pi(\cdot | \zeta)} [\hat{m}(\zeta, a)]$. We have that*

$$|\hat{V}(\pi) - V(\pi)| \leq Y_{\max} \left( \sum_{i=1}^{|\mathcal{X}|} |\hat{p}(x_i) - p(x_i)| \right) + \sup_{\zeta \in \mathcal{X}} \left\{ \left| \hat{l}(\zeta) - l(\zeta) \right| \right\},$$

*where $l(\zeta)$ represents the distributionally robust cost of following policy $\pi$ at the context $\zeta$.*

Lemma A.1 indicates that the gap between $\hat{V}(\pi)$ and $V(\pi)$ is upper bounded by the error from the contexts distribution estimate and the empirical distributionally robust cost function estimate. Since both errors are caused by finite samples, the errors will decrease if we increase the size of the observational dataset.

**Lemma A.2** *Let $P$ be a discrete distribution of size $m$. For each $i \in [m]$, denote $p(i)$ the true probability and $\hat{p}(i) = \frac{1}{n} \sum_{j=1}^{n} \mathbb{I}\{s_j = i\}$ the empirical probability, where $\{s_j\}_{j=1}^{n}$ are $n$ i.i.d. samples drawing from $P$. Denote $f(p) = \sum_{i=1}^{m} |\hat{p}(x_i) - p(x_i)|$ the random distance between $p$ and $\hat{p}$ after observing $n$ samples. For a given $\delta \in (0, 1)$, we have that, with probability at least $1 - \delta$,*

$$f(p) \leq \sqrt{\frac{2 \log (1/\delta)}{n}} + \sqrt{\frac{m}{n}}. \tag{A.1}$$

**Lemma A.3** *Consider a Bernoulli distribution $P$ that takes the value 1 with probability $p > 0$. With probability $1 - \delta$, one observes at least $\frac{np}{2}$ samples of value 1 out of $n$ trials as long as $n \geq \frac{2 \log(1/\delta)}{p^2}$.*

**Lemma A.4** *Let Assumption 2.1 hold. Given $\delta \in (0, 1)$, with probability $1 - \delta$, if we choose $n \geq \frac{2 \log(2|\mathcal{X}||\mathcal{A}|/\delta)}{(\mu_{\pi_0} \mu_{x_0})^2}$, we have $\forall (x, a) \in \mathcal{X} \times \mathcal{A}$, it holds that*

$$|m(x, a) - \hat{m}(x, a)| \leq Y_{max} \left( \sqrt{\frac{4 \log (2|\mathcal{X}||\mathcal{A}|/\delta)}{n \mu_{\pi_0} \mu_{x_0}}} + \sqrt{\frac{2|\Xi|}{n \mu_{\pi_0} \mu_{x_0}}} \right). \tag{A.2}$$

**Proof** [Proof of Theorem 3.2] According to Lemma A.1, we have that

$$|\hat{V}(\pi) - V(\pi)| \leq Y_{\max} \left( \sum_{i=1}^{|\mathcal{X}|} |\hat{p}(x_i) - p(x_i)| \right) + \sup_{\zeta \in \mathcal{X}} \left\{ \left| \hat{l}(\zeta) - l(\zeta) \right| \right\}.$$

We first bound the term $Y_{\max} \left( \sum_{i=1}^{|\mathcal{X}|} |\hat{p}(x_i) - p(x_i)| \right)$, which relates to the total variation between the empirical estimate and the true discrete distribution. For a given $\delta \in (0, 1)$, Lemma A.2 implies that, with probability at least $1 - \delta$, we have that

$$Y_{\max} \left( \sum_{i=1}^{|\mathcal{X}|} |\hat{p}(x_i) - p(x_i)| \right) \leq Y_{\max} \left( \sqrt{\frac{2 \log (1/\delta)}{n}} + \sqrt{\frac{|\mathcal{X}|}{n}} \right), \tag{A.3}$$

where $n$ is the number of samples.

We now bound the term $\sup_{\zeta \in \mathcal{X}} \left\{ \left| \hat{l}(\zeta) - l(\zeta) \right| \right\}$, which relates to the difference between $m(\zeta, a)$ and $\hat{m}(\zeta, a)$ due to the following fact

$$
\begin{aligned}
\sup_{\zeta \in \mathcal{X}} \left\{ \left| \hat{l}(\zeta) - l(\zeta) \right| \right\} &= \sup_{\zeta \in \mathcal{X}} \left\{ \left| \mathbb{E}_{a \sim \pi(\cdot | \zeta)} \left[ \hat{m}(\zeta, a) \right] - \mathbb{E}_{a \sim \pi(\cdot | \zeta)} \left[ m(\zeta, a) \right] \right| \right\} \\
&\leq \sup_{\zeta \in \mathcal{X}, a \in \mathcal{A}} \left\{ \left| \hat{m}(\zeta, a) - m(\zeta, a) \right| \right\}.
\end{aligned}
\tag{A.4}
$$

Lemma A.4 implies that, with probability $1 - \delta$, if we choose $n \geq \frac{2 \log(2|\mathcal{X}||\mathcal{A}|/\delta)}{(\mu_{\pi_0} \mu_{x_0})^2}$, then $|m(x, a) - \hat{m}(x, a)| \leq Y_{\max} \left( \sqrt{\frac{4 \log (2|\mathcal{X}||\mathcal{A}|/\delta)}{n \mu_{\pi_0} \mu_{x_0}}} + \sqrt{\frac{2|\Xi|}{n \mu_{\pi_0} \mu_{x_0}}} \right)$, $\forall (x, a) \in \mathcal{X} \times \mathcal{A}$. Since it holds for all context actions pairs, we have that

$$
\sup_{\zeta \in \mathcal{X}, a \in \mathcal{A}} \left\{ |\hat{m}(\zeta, a) - m(\zeta, a)| \right\} \leq Y_{\max} \left( \sqrt{\frac{4 \log (2|\mathcal{X}||\mathcal{A}|/\delta)}{n \mu_{\pi_0} \mu_{x_0}}} + \sqrt{\frac{2|\Xi|}{n \mu_{\pi_0} \mu_{x_0}}} \right).
\tag{A.5}
$$

Taking a union bound of the events when both inequalities equation A.3 and equation A.5 hold, we have that with probability $1 - \delta$, by selecting $n \geq \frac{2 \log(4|\mathcal{X}||\mathcal{A}|/\delta)}{(\mu_{\pi_0} \mu_{x_0})^2}$, the following inequality holds

$$
|\hat{V}(\pi) - V(\pi)| \leq Y_{\max} \left( \sqrt{\frac{2 \log (2/\delta)}{n}} + \sqrt{\frac{|\mathcal{X}|}{n}} + \sqrt{\frac{4 \log (4|\mathcal{X}||\mathcal{A}|/\delta)}{n \mu_{\pi_0} \mu_{x_0}}} + \sqrt{\frac{2|\Xi|}{n \mu_{\pi_0} \mu_{x_0}}} \right).
$$

∎

## A.2   Proof of Theorem 4.2

**Lemma A.5** *Given a set of $n$ numbers $\{x_1, \cdots, x_n\}$ and let $x_{\max} = \max\{x_1, \cdots, x_n\}$, we define the Log-Sum-Exp (LSE) function as $\mathrm{LSE}(x_1, \cdots, x_n) = \frac{1}{\eta} \log \left( \frac{1}{n} \sum_{i=1}^n \exp(\eta x_i) \right)$. Then, the following inequalities hold:*

$$
x_{\max} - \frac{\log(n)}{\eta} \leq \mathrm{LSE} \left( x_1, \cdots, x_n \right) \leq x_{\max}.
$$

**Proof**   Observe that $\frac{1}{n} \exp(\eta x_{\max}) \leq \frac{1}{n} \sum_{i=1}^n \exp(\eta x_i) \leq \exp(\eta x_{\max})$. The result follows by taking the logarithms to the inequalities and multiplying by $1/\eta$. ∎

Lemma A.5 shows that the DRO problem under the regularized Wasserstein distance can be a good approximation to the original DRO problem if we select the smoothing parameter $\eta$ properly.

**Proof**   Proof of Theorem 4.2:   Lemma A.5 implies that $\frac{1}{\eta} \log \left( \sum_{\zeta \in \mathcal{X}} \frac{1}{|\mathcal{X}|} e^{\eta \left( f(\zeta) - \lambda (x - \zeta)^2 \right)} \right) \leq \sup_{\zeta \in \mathcal{X}} \left( f(\zeta) - \lambda (x - \zeta)^2 \right)$ for any given function $f(\zeta)$. As a result, similar to the proof as in Theorem 3.2, for any given policy $\pi$, both $\hat{V}_\eta(\pi)$ and $\hat{m}_\eta(x, a), \forall (x, a) \in \mathcal{X} \times \mathcal{A}$ are upper bounded by $Y_{\max}$ according to Lemma 2.5. In addition, if we denote $\lambda^*$ and $\hat{\lambda}^*$ the optimal solution to $V$ and $\hat{V}_\eta$, we have $\lambda^*, \hat{\lambda}^* \in [0, Y_{\max}/\epsilon_x]$. Define $l(\zeta) = \mathbb{E}_{a \sim \pi(\cdot | \zeta)} [m(\zeta, a)]$ and $\hat{l}(\zeta) = \mathbb{E}_{a \sim \pi(\cdot | \zeta)} [\hat{m}(\zeta, a)]$. Without loss of generality, for a given policy

$\pi$, we assume $\hat{V}(\pi) \geq V(\pi)$ and we have that

$$|\hat{V}_\eta(\pi) - V(\pi)|$$

$$= \left(\epsilon\hat{\lambda}^* + \sum_{i=1}^{|\mathcal{X}|} \hat{p}(x_i)\frac{1}{\eta}\log\left(\sum_{\zeta\in\mathcal{X}}\frac{1}{|\mathcal{X}|}e^{\eta(\hat{l}(\zeta)-\hat{\lambda}^*(x_i-\zeta)^2)}\right)\right) - \left(\epsilon\lambda^* + \sum_{i=1}^{|\mathcal{X}|} p(x_i)\sup_{\zeta\in\mathcal{X}}\left\{l(\zeta)-\lambda^*(x_i-\zeta)^2\right\}\right)$$

$$\leq \left(\epsilon\lambda^* + \sum_{i=1}^{|\mathcal{X}|} \hat{p}(x_i)\frac{1}{\eta}\log\left(\sum_{\zeta\in\mathcal{X}}\frac{1}{|\mathcal{X}|}e^{\eta(\hat{l}(\zeta)-\lambda^*(x_i-\zeta)^2)}\right)\right) - \left(\epsilon\lambda^* + \sum_{i=1}^{|\mathcal{X}|} p(x_i)\sup_{\zeta\in\mathcal{X}}\left\{l(\zeta)-\lambda^*(x_i-\zeta)^2\right\}\right)$$

$$= \sum_{i=1}^{|\mathcal{X}|}\left(\hat{p}(x_i)\frac{1}{\eta}\log\left(\sum_{\zeta\in\mathcal{X}}\frac{1}{|\mathcal{X}|}e^{\eta(\hat{l}(\zeta)-\lambda^*(x_i-\zeta)^2)}\right) - p(x_i)\sup_{\zeta\in\mathcal{X}}\left\{l(\zeta)-\lambda^*(x_i-\zeta)^2\right\}\right),$$

where the inequality holds since $\hat{\lambda}^*$ is the optimal solution to $\hat{V}_\eta(\pi)$ while $\lambda^*$ is not. Then, it follows that

$$|\hat{V}_\eta(\pi) - V(\pi)|$$

$$\leq \sum_{i=1}^{|\mathcal{X}|}\left(\hat{p}(x_i)\frac{1}{\eta}\log\left(\sum_{\zeta\in\mathcal{X}}\frac{1}{|\mathcal{X}|}e^{\eta(\hat{l}(\zeta)-\lambda^*(x_i-\zeta)^2)}\right) - p(x_i)\frac{1}{\eta}\log\left(\sum_{\zeta\in\mathcal{X}}\frac{1}{|\mathcal{X}|}e^{\eta(\hat{l}(\zeta)-\lambda^*(x_i-\zeta)^2)}\right)\right.$$

$$+ p(x_i)\frac{1}{\eta}\log\left(\sum_{\zeta\in\mathcal{X}}\frac{1}{|\mathcal{X}|}e^{\eta(\hat{l}(\zeta)-\lambda^*(x_i-\zeta)^2)}\right) - p(x_i)\sup_{\zeta\in\mathcal{X}}\left\{\hat{l}(\zeta)-\lambda^*(x_i-\zeta)^2\right\}$$

$$\left. + p(x_i)\sup_{\zeta\in\mathcal{X}}\left\{\hat{l}(\zeta)-\lambda^*(x_i-\zeta)^2\right\} - p(x_i)\sup_{\zeta\in\mathcal{X}}\left\{l(\zeta)-\lambda^*(x_i-\zeta)^2\right\}\right)$$

$$\leq Y_{\max}\sum_{i=1}^{|\mathcal{X}|}|\hat{p}(x_i)-p(x_i)| + \frac{\log(|\mathcal{X}|)}{\eta} + \sum_{i=1}^{|\mathcal{X}|}p(x_i)\sup_{\zeta\in\mathcal{X}}\left\{\left|\hat{l}(\zeta)-l(\zeta)\right|\right\},$$

The last inequality holds since $\frac{1}{\eta}\log\left(\sum_{\zeta\in\mathcal{X}}\frac{1}{|\mathcal{X}|}e^{\eta(\hat{l}(\zeta)-\lambda^*(x_i-\zeta)^2)}\right) \leq \max_{\zeta\in\mathcal{X}}\{\hat{l}(\zeta) - \lambda^*(x_i - \zeta)^2\} \leq \max_{\zeta\in\mathcal{X}}\{\hat{l}(\zeta)\} \leq Y_{\max}$. In addition, according to Lemma A.5, we have that $\left|\frac{1}{\eta}\log\left(\sum_{\zeta\in\mathcal{X}}\frac{1}{|\mathcal{X}|}e^{\eta(\hat{l}(\zeta)-\lambda^*(x_i-\zeta)^2)}\right) - \sup_{\zeta\in\mathcal{X}}\left\{\hat{l}(\zeta)-\lambda^*(x_i-\zeta)^2\right\}\right| \leq \frac{\log(|\mathcal{X}|)}{\eta}$. Observe that the last inequality is the same as in Lemma A.1 except for the extra error term $\frac{\log(|\mathcal{X}|)}{\eta}$; we complete the proof by applying the results from Theorem 3.2 to the last inequality. ∎

## B    Proofs of Supporting Lemmas

### B.1    Proof of Lemma 2.5

**Proof**    Denote $D^* = \epsilon\lambda^* + \mathbb{E}_{x\sim P^0}\left[\sup_{\zeta\in\mathcal{X}}\left(f(\zeta)-\lambda^*(x-\zeta)^2\right)\right]$. For a given point $x$ and $\lambda^*$, we have that $\sup_{\zeta\in\mathcal{X}}\left(f(\zeta)-\lambda^*(x-\zeta)^2\right) \geq \sup_{\zeta=x}\left(f(\zeta)-\lambda^*(x-\zeta)^2\right)$. Taking the expectation of $x$ with respect to the distribution $P^0$ and adding $\epsilon\lambda^*$ on both sides, we have $\epsilon\lambda^* + \mathbb{E}_{x\sim P^0}\left[\sup_{\zeta\in\mathcal{X}}\left(f(\zeta)-\lambda^*(x-\zeta)^2\right)\right] \geq \epsilon\lambda^* + \mathbb{E}_{x\sim P^0}\left[\sup_{\zeta=x}\left(f(\zeta)-\lambda^*(x-\zeta)^2\right)\right] = \epsilon\lambda^* + \mathbb{E}_{x\sim P^0}\left[f(x)\right]$, which implies $D^* \geq \epsilon\lambda^* + \mathbb{E}_{x\sim P^0}\left[f(x)\right] \geq \epsilon\lambda^*$. By the definition of the dual problem, we have $D^* \leq \epsilon\cdot 0 + \mathbb{E}_{x\sim P^0}\left[\sup_{\zeta\in\mathcal{X}}\left(f(\zeta)-0\cdot(x-\zeta)^2\right)\right] = f_{\max}$. The result follows from the fact that $0 \leq \epsilon\lambda^* \leq D^* \leq f_{\max}$. ∎

## B.2 Proof of Lemma A.1

**Proof** Denote $\lambda^*$ and $\hat{\lambda}^*$ the optimal solution to $V$ and $\hat{V}$. Applying Lemma 2.5 to equation 3.2 and equation 3.4, we have $m(s,a)$ and $\hat{m}(s,a)$ are upper bounded by $Y_{\max}$ since the costs are upper bounded by $Y_{\max}$. This further implies that $l(x)$ and $\hat{l}(x)$ are upper bounded by $Y_{\max}$. Applying Lemma 2.5 to equation 3.1 and equation 3.3, we have $\lambda^*, \hat{\lambda}^* \in [0, Y_{\max}/\epsilon_x]$. Without loss of generality, for a given policy $\pi$, we assume $\hat{V}(\pi) \geq V(\pi)$ and we have $|\hat{V}(\pi) - V(\pi)|$

$$
= (\epsilon\hat{\lambda}^* + \sum_{i=1}^{|\mathcal{X}|} \hat{p}(x_i) \sup_{\zeta \in \mathcal{X}} \left\{\hat{l}(\zeta) - \hat{\lambda}^*(x_i - \zeta)^2\right\}) - (\epsilon\lambda^* + \sum_{i=1}^{|\mathcal{X}|} p(x_i) \sup_{\zeta \in \mathcal{X}} \left\{l(\zeta) - \lambda^*(x_i - \zeta)^2\right\})
$$

$$
\leq (\epsilon\lambda^* + \sum_{i=1}^{|\mathcal{X}|} \hat{p}(x_i) \sup_{\zeta \in \mathcal{X}} \left\{\hat{l}(\zeta) - \lambda^*(x_i - \zeta)^2\right\}) - (\epsilon\lambda^* + \sum_{i=1}^{|\mathcal{X}|} p(x_i) \sup_{\zeta \in \mathcal{X}} \left\{l(\zeta) - \lambda^*(x_i - \zeta)^2\right\})
$$

$$
= \sum_{i=1}^{|\mathcal{X}|} \left( \hat{p}(x_i) \sup_{\zeta \in \mathcal{X}} \left\{\hat{l}(\zeta) - \lambda^*(x_i - \zeta)^2\right\} - p(x_i) \sup_{\zeta \in \mathcal{X}} \left\{l(\zeta) - \lambda^*(x_i - \zeta)^2\right\} \right).
$$

The first inequality follows from the fact $\hat{\lambda}^*$ is the optimal solution to $\hat{V}$ while $\lambda^*$ is not. Then, we have

$$
|\hat{V}(\pi) - V(\pi)| \leq \sum_{i=1}^{|\mathcal{X}|} \left( \hat{p}(x_i) \sup_{\zeta \in \mathcal{X}} \left\{\hat{l}(\zeta) - \lambda^*(x_i - \zeta)^2\right\} - p(x_i) \sup_{\zeta \in \mathcal{X}} \left\{\hat{l}(\zeta) - \lambda^*(x_i - \zeta)^2\right\} \right.
$$

$$
\left. + p(x_i) \sup_{\zeta \in \mathcal{X}} \left\{\hat{l}(\zeta) - \lambda^*(x_i - \zeta)^2\right\} - p(x_i) \sup_{\zeta \in \mathcal{X}} \left\{l(\zeta) - \lambda^*(x_i - \zeta)^2\right\} \right)
$$

$$
\leq \sum_{i=1}^{|\mathcal{X}|} |\hat{p}(x_i) - p(x_i)| \sup_{x,\zeta \in \mathcal{X}} \left\{\hat{l}(\zeta) - \lambda^*(x - \zeta)^2\right\} + \sum_{i=1}^{|\mathcal{X}|} p(x_i) \sup_{\zeta \in \mathcal{X}} \left\{\left|\hat{l}(\zeta) - l(\zeta)\right|\right\}
$$

$$
\leq Y_{\max} \left( \sum_{i=1}^{|\mathcal{X}|} |\hat{p}(x_i) - p(x_i)| \right) + \sum_{i=1}^{|\mathcal{X}|} p(x_i) \sup_{\zeta \in \mathcal{X}} \left\{\left|\hat{l}(\zeta) - l(\zeta)\right|\right\}.
$$

The second inequality holds since $\sup_x f(x) - \sup_x g(x) \leq \sup_x |f(x) - g(x)|$. The last inequality holds due to the fact that $\hat{l}(\zeta) \leq Y_{\max}$ and $\lambda^* \geq 0$. Note that if $\hat{V}(\pi) \leq V(\pi)$, we will replace the term $\sup_{x,\zeta \in \mathcal{X}} \left\{\hat{l}(\zeta) - \lambda^*(x - \zeta)^2\right\}$ by the term $\sup_{x,\zeta \in \mathcal{X}} \left\{l(\zeta) - \hat{\lambda}^*(x - \zeta)^2\right\}$ in first inequality and the result still holds. ∎

## B.3 Proof of Lemma A.2

**Proof** Observe that changing any single sample can at most change the value of $f$ by $1/n$. Applying McDiarmid's inequality to the function $f$, we have that

$$
\mathbb{P}\left[f(p) - \mathbb{E}[f(p)] \geq \epsilon\right] \leq e^{-\frac{2\epsilon^2}{n(\frac{1}{n})^2}} = e^{-2n\epsilon^2},
$$

which implies with probability at least $1 - \delta$, $f(p) - \mathbb{E}[f(p)] \leq \sqrt{\frac{2\log(1/\delta)}{n}}$ by setting $\epsilon = \sqrt{\frac{2\log(1/\delta)}{n}}$. Thus, to bound $f(p)$, we only need to bound the first-moment term $\mathbb{E}[f(p)]$. Specifically, we have that

$$
\mathbb{E}[f(p)] = \sum_{i=1}^{m} \mathbb{E}\left[|\hat{p}(x_i) - p(x_i)|\right] \leq \sum_{i=1}^{m} \sqrt{\mathbb{E}\left[|\hat{p}(x_i) - p(x_i)|^2\right]} = \sum_{i=1}^{m} \sqrt{\frac{1}{n} p(x_i) \left(1 - p(x_i)\right)},
$$

where the first inequality follows from Jensen's inequality. The second inequality holds due to the fact that $n\hat{p}(x_i)$ follows from the binomial distribution and $\mathbb{E}\left[|\hat{p}(x_i) - p(x_i)|^2\right] = 1/n^2 \cdot \text{Var}(n\hat{p}(x_i)) = 1/n^2 \cdot np(x_i)(1-p(x_i))$. Since $1 - p(x_i) \leq 1$, we have that

$$\sum_{i=1}^{m} \sqrt{\frac{1}{n} p(x_i)\left(1 - p(x_i)\right)} \leq \sum_{i=1}^{m} \sqrt{\frac{1}{n}} \sqrt{p(x_i)} \leq \sqrt{\frac{1}{n}} \sqrt{m} = \sqrt{\frac{m}{n}},$$

where the second inequality follows from Cauchy–Schwarz inequality. Putting everything together, we have $f(p) \leq \sqrt{\frac{2 \log(1/\delta)}{n}} + \sqrt{\frac{m}{n}}$. ∎

## B.4 Proof of Lemma A.3

**Proof** Let $\hat{p}$ the empirical estimate of $p$ after $n$ trials, i.e., $\hat{p} = \frac{1}{n} \sum_{i=1}^{n} \mathbb{I}\{s_j = 1\}$, where $\{s_j\}_{j=1}^{n}$ are $n$ i.i.d. samples drawing from $P$. We first show that the probability $\hat{p} \leq \frac{p}{2}$ is low. We have that

$$\mathbb{P}\left(\hat{p} \leq \frac{p}{2}\right) = \mathbb{P}\left((p - \hat{p}) \geq \frac{p}{2}\right) \leq \exp\left(-2n\left(\frac{p}{2}\right)^2\right) = \exp\left(-\frac{np^2}{2}\right),$$

where the inequality follows from Hoeffding's inequality. For a given $\delta > 0$, by choosing $n \geq \frac{2\log(1/\delta)}{p^2}$, we have that with probability at least $1 - \delta$, $\hat{p} \geq \frac{p}{2}$. This implies that we observe at least $\frac{np}{2}$ samples of value 1 out of $n$ trials. ∎

## B.5 Proof of Lemma A.4

**Proof** Without loss of generality, we assume $\hat{m}(x, a) \geq m(x, a)$ and let $\lambda^*$ and $\hat{\lambda}^*$ the optimal solution to $m$ and $\hat{m}$, we have that

$$|\hat{m}(x, a) - m(x, a)|$$

$$= \left(\epsilon\hat{\lambda}^* + \mathbb{E}_{\xi \sim \hat{P}_{\Xi}^0} \sup_{\zeta \in \Xi} \left(y_{x,a}(\zeta) - \hat{\lambda}^*(\xi - \zeta)^2\right)\right) - \left(\epsilon\lambda^* + \mathbb{E}_{\xi \sim P_{\Xi_0}} \sup_{\zeta \in \Xi} \left(y_{x,a}(\zeta) - \lambda^*(\xi - \zeta)^2\right)\right)$$

$$\leq \left(\epsilon\lambda^* + \mathbb{E}_{\xi \sim \hat{P}_{\Xi}^0} \sup_{\zeta \in \Xi} \left(y_{x,a}(\zeta) - \lambda^*(\xi - \zeta)^2\right)\right) - \left(\epsilon\lambda^* + \mathbb{E}_{\xi \sim P_{\Xi_0}} \sup_{\zeta \in \Xi} \left(y_{x,a}(\zeta) - \lambda^*(\xi - \zeta)^2\right)\right)$$

$$\leq Y_{\max} \sum_{i=1}^{|\Xi|} |\hat{p}(\xi_i) - p(\xi_i)|.$$

Observe that $\sum_{i=1}^{|\Xi|} |\hat{p}(\xi_i) - p(\xi_i)|$ directly depends on the number of samples. Denote $n(x, a)$ the random variable representing the number of $(x, a)$ pairs observed in the dataset. Let $p_{x,a}$ be the probability of observing each pair $(x, a)$. Lemma A.3 implies that if we select $n \geq \frac{2\log(1/\delta)}{p_{x,a}^2}$, we will observe at least $\frac{np_{x,a}}{2}$ samples of $(s, a)$ pairs with high probability. In addition, according to Lemma A.2, with high probability, we have

$$\sum_{i=1}^{|\Xi|} |\hat{p}(\xi_i) - p(\xi_i)| \leq \sqrt{\frac{2 \log(1/\delta)}{n(x, a)}} + \sqrt{\frac{|\Xi|}{n(x, a)}}$$

Take a union bound on the following two events

$$\left\{n(x, a) \geq \frac{np_{x,a}}{2}\right\} \text{ and } \left\{\sum_{i=1}^{|\Xi|} |\hat{p}(\xi_i) - p(\xi_i)| \leq \sqrt{\frac{2 \log(1/\delta)}{n(s, a)}} + \sqrt{\frac{|\Xi|}{n(x, a)}}\right\},$$

we have that, with probability $1 - \delta$, if we select $n \geq \frac{2\log(2/\delta)}{p_{x,a}^2}$, then

$$\sum_{i=1}^{|\Xi|} |\hat{p}(\xi_i) - p(\xi_i)| \leq \sqrt{\frac{4\log(2/\delta)}{np_{x,a}}} + \sqrt{\frac{2|\Xi|}{np_{x,a}}}.$$

We complete the proof by taking the union bounds over all state action pairs and using the fact that $p_{x,a}$ is lower bounded by $\mu_{\pi_0}\mu_{x_0}$, which is implied by Assumption 2.1. ∎

## C  Proofs for Off-Policy Learning

### C.1  Proof of Theorem 5.1

**Proof**  When $\hat{V}^* \geq V^*$, we obtain that $|\hat{V}^* - V^*| = \hat{V}(\hat{\pi}^*) - V(\pi^*) \leq \hat{V}(\pi^*) - V(\pi^*)$. Then, the result follows by applying Theorem 3.2 to the policy $\pi^*$. When $V^* \geq \hat{V}^*$, we have that $|\hat{V}^* - V^*| = V(\pi^*) - \hat{V}(\hat{\pi}^*) \leq V(\hat{\pi}^*) - \hat{V}(\hat{\pi}^*)$. The result follows by applying Theorem 3.2 to the policy $\hat{\pi}^*$. ∎

### C.2  Proof of Theorem 5.2

Due to page limit, the sample size and the learning rate are not specified in Theorem 5.2. The theorem below provides a complete version of Theorem 5.2.

**Theorem C.1** *(Query complexity of Algorithm 1). Assume that the policy space $\Theta$ is compact and convex and there exist a finite number $B > 0$ such that for any $x, y \in \mathcal{X}$, $\|x - y\|_2 \leq B$. i) Suppose that $\hat{l}(\theta, \zeta) = \sum_{i=1}^{k} \pi_\theta(a_i|\zeta)\hat{m}(\zeta, a_i)$ is convex in $\theta$ and $L_{\hat{l}}$-smooth in $\theta$ for any $\zeta \in \mathcal{X}$. For a given accuracy $\delta > 0$, if we set $m_t = \mathcal{O}(\delta^{-1})$ and $\gamma_t = \mathcal{O}(T^{-1/2})$, then it requires $\mathcal{O}(\delta^{-2})$ iterations to find a $\delta$-accurate optimal solution. The total query complexity is $\mathcal{O}(\delta^{-3})$. ii) Suppose that $\hat{l}(\theta, \zeta) = \sum_{i=1}^{k} \pi_\theta(a_i|\zeta)\hat{m}(\zeta, a_i)$ is $L_{\hat{l}}$-smooth in $\theta$ for any $\zeta \in \mathcal{X}$. For a given accuracy $\delta > 0$, if we set $m_t = \mathcal{O}(\delta^{-2})$ and $\gamma_t = \mathcal{O}(T^{-1/2})$, then it requires $\mathcal{O}(\delta^{-4})$ iterations to find a $\delta$-accurate stationary point. The total query complexity is $\mathcal{O}(\delta^{-6})$.*

Before proving Theorem C.1, we first introduce some notations and study our problem under the framework of conditional stochastic optimization (Hu et al., 2020).

**Notations**  A function $f(\cdot) : \mathbb{R}^k \rightarrow \mathbb{R}$ is said to be $L$-Lipschitz continuous on $\mathcal{X}$ if for any $x, y \in \mathcal{X}$, $|f(x) - f(y)| \leq L\|x - y\|_2$; $f(\cdot)$ is said to be $S$-Lipschitz smooth on $\mathcal{X}$ if for any $x, y \in \mathcal{X}$, $f(x) - f(y) - \nabla f(y)^\top(x-y) \leq \frac{S}{2}\|x-y\|_2^2$; $f(\cdot)$ is said to be $\mu$-convex on $\mathcal{X}$ if for any $x, y \in \mathcal{X}$, $f(x) - f(y) - \nabla f(y)^\top(x-y) \geq \frac{\mu}{2}\|x - y\|_2^2$.

**Conditional stochastic optimization**  Consider the following conditional stochastic optimization problem:

$$\min_{u \in \mathcal{U}} F(u) = \mathbb{E}_x \left[ f_x \left( \mathbb{E}_{\zeta|x} \left[ g_\zeta(u, x) \right] \right) \right], \tag{C.1}$$

where $\mathcal{U} \subset \mathbb{R}^d, g_\zeta(\cdot, x)$ is a function that depends on random variables $x$ and $\zeta$. For a fixed $x$ and given $m$ samples $\{\zeta_j\}_{j=1}^m$, define the finite sample function

$$\hat{F}(u; x, \{\zeta_j\}_{j=1}^m) = f_x \left( \frac{1}{m} \sum_{j=1}^{m} g_{\zeta_j}(u, x) \right).$$

In addition, the biased gradient of $F(u)$ is defined as:

$$\nabla_u \hat{F}(u; x, \{\zeta_j\}_{j=1}^m) = \left(\nabla_u f_x \left(\frac{1}{m}\sum_{j=1}^m g_{\zeta_j}(u,x)\right)\right)^\top \left(\frac{1}{m}\sum_{j=1}^m \nabla_u g_{\zeta_j}(u,x)\right).$$

Recall the regularized Wasserstein DRO problem is defined as:

$$\hat{V}_\eta(\pi^*) = \inf_{\theta, \lambda \geq 0}\left\{\mathbb{E}_{x \sim \hat{P}_x^0}\left[\frac{1}{\eta}\log\left(\sum_{\zeta \in \mathcal{X}}\frac{1}{|\mathcal{X}|}e^{\eta\left(\hat{l}(\theta,\zeta)-\lambda(x-\zeta)^2\right)}\right)\right] + \epsilon_x \lambda\right\}. \tag{C.2}$$

We now convert the regularized Wasserstein DRO problem into the conditional stochastic optimization problem. Denote $u = (\theta, \lambda)$, for a fixed $x$ sample from $\hat{P}_x^0$, we define

$$g_\zeta(u,x) = e^{\eta\left(\hat{l}(\theta,\zeta)-\lambda(x-\zeta)^2\right)} \cdot e^{\eta \epsilon_x \lambda}, \quad f_x(\cdot) = \frac{1}{\eta}\log(\cdot).$$

Thus we have $\hat{V}_\eta(\pi^*) = \inf_u \hat{F}(u) = \inf_u \left\{\mathbb{E}_{x \sim \hat{P}_x^0}\left[f_x\left(\mathbb{E}_{\zeta \sim \text{uniform}(\mathcal{X})}\left[g_\zeta(u,x)\right]\right)\right]\right\}$. The regularized Wasserstein DRO problem is indeed a conditional stochastic optimization problem. Its biased gradient with respect to $\theta, \lambda$ can be computed as follows:

$$\nabla_\theta \hat{F}(\theta, \lambda; x, \{\zeta_j\}_{j=1}^m) = \frac{\sum_{j=1}^m e^{\eta\left(\hat{l}(\theta,\zeta_j)-\lambda(x-\zeta_j)^2\right)}\nabla_\theta \hat{l}(\theta,\zeta_j)}{\sum_{j=1}^m e^{\eta\left(\hat{l}(\theta,\zeta_j)-\lambda(x-\zeta_j)^2\right)}},$$

$$\nabla_\lambda \hat{F}(\theta, \lambda; x, \{\zeta_j\}_{j=1}^m) = -\frac{\sum_{j=1}^m e^{\eta\left(\hat{l}(\theta,\zeta_j)-\lambda(x-\zeta_j)^2\right)}(x-\zeta_j)^2}{\sum_{j=1}^m e^{\eta\left(\hat{l}(\theta,\zeta_j)-\lambda(x-\zeta_j)^2\right)}} + \epsilon_x.$$

As proven in Theorem 3.2 and 3.3 in Hu et al. (2020), to prove Theorem C.1, it suffices to show that the following properties hold for our problem:

1. Bounded variance: $\sigma_g^2 := \sup_{x,u \in \mathcal{U}} \mathbb{E}_{\zeta|x}\|g_\zeta(u,x) - \mathbb{E}_{\zeta|x}[g_\zeta(u,x)]\|_2^2 < +\infty$

2. $\hat{F}(u; x, \{\zeta_j\}_{j=1}^m)$ is convex in $u$ if $\hat{l}(\theta, \zeta)$ is convex in $\theta$ and is $L_{\hat{l}}$-smooth in $\theta$ (convex case); $f_x$ and $g_{z_j}$ are Lipschitz continuous and smooth if $\hat{l}(\theta, \zeta)$ is only $L_{\hat{l}}$-smooth in $\theta$ (non-convex case)

3. Bounded second moment: there exist $M > 0$ such that $\mathbb{E}\left[\|\nabla_u \hat{F}(u; x, \{\zeta_j\}_{j=1}^m)\|_2^2|x\right] \leq M^2$ for any $u$.

1. We first verify that regularized DRO problem satisfies the bounded variance propriety. Note that for any $x, u, \zeta$, as $0 \leq \hat{l} \leq Y_{max}$ and $0 \leq \lambda \leq Y_{max}/\epsilon_x$, we have that

$$e^{\eta\left(0 - \frac{Y_{\max}}{\epsilon_x}B^2 + 0\right)} \leq g_\zeta(x,u) \leq e^{\eta\left(Y_{\max} - 0 + \epsilon_x \frac{Y_{\max}}{\epsilon_x}\right)},$$

which implies that $\sigma_g^2 \leq \left(e^{2\eta Y_{\max}} - e^{-\eta Y_{\max}B^2/\epsilon_x}\right)^2 \leq e^{4\eta Y_{\max}} < +\infty$.

2. We now verify the convex case of the second propriety. For a given $x$ and $\zeta_j$, denote $h(\theta, \lambda; x, \zeta_j) = \hat{l}(\theta, \zeta_j) - \lambda(x - \zeta_j)^2$. As $\hat{l}(\theta, \zeta_j)$ is convex in $\theta$ by assumption and $\lambda(x - \zeta_j)^2$ is linear in $\lambda$, $h(\theta, \lambda; x, \zeta_j)$ is convex in $\theta$ and $\lambda$ jointly. Consequently, for a given $x \in \mathcal{X}$ and $\{\zeta_j\}_{j=1}^m$, the function $\hat{F}(u; x, \{\zeta_j\}_{j=1}^m) = \frac{1}{\eta}\log\left(\sum_{j=1}^m \frac{1}{m}e^{\eta\left(\hat{l}(\theta,\zeta_j)-\lambda(x-\zeta_j)^2\right)}\right) + \epsilon_x \lambda$ is convex in $(\theta, \lambda)$ since $\log\sum_{j=1}^m \frac{1}{m}\exp\left(h(\theta, \lambda; x, \zeta_j)\right)$ preserves convexity when $h(\theta, \lambda; x, \zeta_j)$ is convex in $(\theta, \lambda)$.

For the non-convex case, we need to show both $f_x$ and $g_{\zeta_j}$ are smooth. As derived in the first property, $g_\zeta$ is bounded. Thus, the domain of the function $f_x$ is restricted to the set $[e^{-\eta Y_{\max}B^2/\epsilon_x}, e^{2\eta Y_{\max}}]$ and thus

Table 2: Comparison between KL DRO and Wasserstein DRO

| Method | $0.8\epsilon_1^*$ | $\epsilon_1^*$ | $1.2\epsilon_1^*$ | $0.8\epsilon_2^*$ | $\epsilon_2^*$ | $1.2\epsilon_2^*$ |
|---|---|---|---|---|---|---|
| KL DRO | 47.27 | 49.84 | 52.20 | 49.86 | 53.07 | 56.09 |
| Wasserstein DRO | 41.53 | 43.36 | 45.04 | 41.82 | 43.72 | 45.49 |
| Expectation under $Q$ | 35.58 | | | | | |

$f_x(\cdot) = \frac{1}{\eta}\log(\cdot)$ is $e^{\eta Y_{\max}B^2/\epsilon_x}/\eta$-Lipschitz continuous and $e^{2\eta Y_{\max}B^2/\epsilon_x}/\eta^2$-Lipschitz smooth. We now show that $g_\zeta(\cdot, x)$ is smooth. For any given two points $u$ and $u'$, we have that

$$\|\nabla_u g_\zeta(u,x) - \nabla_u g_\zeta(u',x)\| \leq \|\nabla_\theta g_\zeta(u,x) - \nabla_\theta g_\zeta(u',x)\| + \|\nabla_\lambda g_\zeta(u,x) - \nabla_\lambda g_\zeta(u',x)\|$$

$$= \|g_\zeta(u,x)\eta\nabla_\theta\hat{l}(\theta,\zeta) - g_\zeta(u',x)\eta\nabla_\theta\hat{l}(\theta',\zeta)\| + \|(g_\zeta(u,x) - g_\zeta(u',x))\,\eta(\epsilon_x - (x-\zeta)^2)\|, \tag{C.3}$$

where the equality follows from the definition of $g_\zeta$ functions for the regularized DRO problem. As $\hat{l}$ is Lipschitz smooth in $\theta$, we can further bound equation C.3 as follows

$$\text{equation } C.3 \leq \eta e^{2\eta Y_{\max}}\|\nabla_\theta\hat{l}(\theta,\zeta) - \nabla_\theta\hat{l}(\theta',\zeta)\| + \eta B^2\|g_\zeta(u,x) - g_\zeta(u',x)\|$$

$$\leq \eta e^{2\eta Y_{\max}}L_{\hat{l}}\|\theta - \theta'\| + \eta B^2 e^{\eta Y_{\max}}\|e^{\eta\epsilon_x\lambda} - e^{\eta\epsilon_x\lambda'}\|$$

$$\leq \eta e^{2\eta Y_{\max}}L_{\hat{l}}\|\theta - \theta'\| + \eta B^2 e^{2\eta Y_{\max}}\|\lambda - \lambda'\| \leq \sqrt{2}\eta(B^2+1)e^{2\eta Y_{\max}}(L_{\hat{l}}+1)\|u - u'\|.$$

As a result, $g_\zeta(u,x)$ is $\sqrt{2}\eta(B^2+1)e^{2\eta Y_{\max}}(L_{\hat{l}}+1)$-Lipschitz smooth in $u$ for any $x,\zeta$.

3. The bounded second moment property hold due to the fact that both $\|\nabla_\theta\hat{l}(\theta,\zeta_j)\|$ and $\|x - \zeta_j\|$ are bounded as $\Theta$ is compact and $\|x - \zeta_j\| \leq B$.

As all three properties hold for the regularized Wasserstein DRO problem, we complete the proof.

## D Connection of regularized Wasserstein DRO to entropy regularized primal problems

**Lemma D.1** *(Strong duality for cost-regularized Wasserstein DRO (Azizian et al., 2022, Theorem 3.1)). Take $\epsilon, \eta > 0$. Then we have*

$$\sup_{P\in\mathcal{U}(\epsilon;P_0)}\mathbb{E}_{\xi\sim P}[f(\xi)] - \frac{1}{\eta}\mathrm{KL}(\sigma|\sigma_0)\inf_{\lambda\geq 0}\left\{\epsilon\lambda + \mathbb{E}_{x\sim P_0}\left[\frac{1}{\eta}\log\left(\mathbb{E}_{\zeta\sim\sigma_0(\cdot|x)}\left[e^{\eta(f(\zeta)-\lambda(x-\zeta)^2)}\right]\right)\right]\right\},$$

*where $\mathcal{U}(\epsilon;P_0) = \{P\in\mathcal{M}^+ : \inf_\sigma\{\mathbb{E}_{(\xi,\zeta)\sim\sigma}[c(\xi,\zeta)] : \sigma_1 = P^0, \sigma_2 = P\} \leq \epsilon\}$ and $\sigma_0(\cdot|x)$ is a pre-specified probability measure for each $x\in\mathcal{X}$.*

The dual problem in Lemma D.1 coincides with our dual formulation equation 4.1 if $\sigma_0(\cdot|x)$ is chosen as a uniform distribution over $\mathcal{X}$.

**Remark D.2** *Note that the entropy-regularized term in the primal problem is added to the objective function. It is possible to add the entropy-regularized term on the uncertainty set as well, which is referred to as Sinkhorn distance in the optimal transport literature (Cuturi, 2013). Distributionally robust optimization using Sinkhorn distance has recently been explored in Wang et al. (2021). Our construction is motivated by directly smoothing the dual problem while Sinkhorn DRO starts from regularizing the primal problem. A similar idea of smoothing dual problems was studied for semi-supervised learning problems; see Blanchet & Kang (2020) for details.*

## E Comparison between KL DRO and Wasserstein DRO

We provide a simple DRO example to illustrate the differences when using different uncertainty set measures. Let $\mathcal{X} = \{x_i\}_{i=1}^{51}$ be the support set that spanning uniformly from 0 to 10. The true training distribution

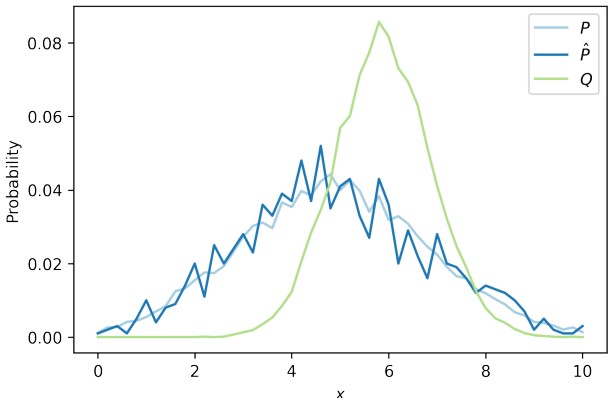

Figure 2: Distributions

$P$, the empirical training distribution $\hat{P}$ and testing distribution $Q$ are depicted in Figure 2. Note that only the empirical distribution $\hat{P}$ can be observed. We define the cost function $f : \mathcal{X} \to \mathbb{R} = x^2$. We then set the radius for both KL ($\epsilon^*_{\text{KL}}$) and Wasserstein ($\epsilon^*_{\text{W}}$) by directly calculating the distance between $\hat{P}$ and $Q$. The robust values are shown in Table 2. With a slightly abuse of notation, $\epsilon^*_1$ is defined as $\text{KL}(Q||\hat{P}) = 0.52$ and $\text{W}(\hat{P}, Q) = 1.85$ for each method, respectively. Since in practice we don't observe $Q$ and cannot compute the true distance $\epsilon^*$, we also provide results when $\epsilon$ is not accurate, e.g., when $\epsilon = 0.8\epsilon^*_1$ or $\epsilon = 1.2\epsilon^*_1$. The expectation under $Q$ method represents the true expected cost when $x$ is distributed according to the testing distribution $Q$. Both DRO methods provide valid upper bounds to the true expected costs. As mentioned in Section 2, KL DRO tends to over-fit the largest cost and is not aware of the geometry of the support. Here, we modify the distribution $\hat{P}$ by setting $x_{51} = 12$ and the cost $f(x_{51}) = x^2_{51}$ and keep the probability $\hat{P}(X = x_{51})$ the same as before, i.e., only the last point of the support is modified. We call the modified distribution $\tilde{P}$. This new point $x_{51}$ can be seen as an outlier to the original support set. We then define $\epsilon^*_2 = \text{KL}(Q||\tilde{P}) = 0.52$ and $\epsilon^*_2 = \text{W}(\tilde{P}, Q) = 1.92$ for each method, respectively. Observe that the KL divergence does not change since the probability $\hat{P}(X = x_{51})$ remains the same while the Wasserstein distance increases as we increase one value of the support. As shown in Table 2, the Wasserstein DRO method is robust to the outlier; the robust values obtained by the KL DRO method increase substantially though the true expect cost remains the same.

In practice, if the dataset is not free of outliers and the outliers incur larger costs, it is better to use the Wasserstein DRO method since it considers the geometry of the dataset and does not simply over-fit to high costs.

# F  Additional Details of the Stroke Trial Application

The selected contextual variables, actions and follow-up events are described in Table 3. The cost function is defined according to the random follow-up events:

$$\text{cost} = \mathbb{I}\{\text{ISC14}\} + \mathbb{I}\{\text{NK14}\} + \mathbb{I}\{\text{HTI14}\} + \mathbb{I}\{\text{PE14}\}$$
$$+ \mathbb{I}\{\text{DVT14}\} + \mathbb{I}\{\text{NCB14}\} + 3 \cdot \mathbb{I}\{\text{DDEAD}\}.$$

After removing patients according to the selection bias, the training set contains 4,340 patients. However, since there are 9,936 distinct contexts, the number of data points for most contexts is limited. To address this issue, we train two decision tree models , one for each action, to learn the average cost of each context and action pair by grouping similar contexts together. We further group the contextual variables AGE and RSBP into bins with size of 10 to reduce the support sizes. The decision trees parameters are cross-validated and shown in Table 4. The same decision trees are used as inputs to all KL and Wasserstein DRO OPE and OPL methods that are listed in Table 1.

Table 3: Dataset summary

| Variables | Descriptions |
|---|---|
| RCONSC | Conscious state at randomization (F-fully alert, D-drowsy, U-unconscious) |
| SEX | M = male; F = female |
| AGE | Age in years |
| RSLEEP | Symptoms noted on waking (Y/N) |
| RATRIAL | Atrial fibrillation (Y/N) |
| RSBP | Systolic blood pressure at randomisation (mmHg) |
| STYPE | Stroke subtype (TACS/PACS/POCS/LACS) |
| Action 1 | If aspirin is allocated and heparin is allocated either 12500 or 5000 unites |
| Action 2 | If both aspirin and heparin are not allocated |
| DDEAD | Dead on discharge form within 14 days (Y/N) |
| ISC14 | Indicator of ischaemic stroke within 14 days (Y/N) |
| NK14 | Indicator of indeterminate stroke within 14 days (Y/N) |
| HTI14 | Indicator of haemorrhagic transformation within 14 days (Y/N) |
| PE14 | Indicator of pulmonary embolism within 14 days (Y/N) |
| DVT14 | Indicator of deep vein thrombosis on discharge form (Y/N) |
| NCB14 | Indicator of any non-cerebral bleed within 14 days (Y/N) |

Table 4: Decision trees parameters

| Parameters | Action 1 | Action 2 |
|---|---|---|
| max depth | 4 | 4 |
| min samples leaf | 5 | 2 |
| score function | mean squared error | mean squared error |

We calculate the initial Wasserstein distance $\epsilon_w = 0.03$ by solving a linear program. Since the number of optimization variables in the linear program is very large (approximately $10^8$ variables), affecting our ability to solve it, we restrict linear program to the subset of the support $\mathcal{X}$, specifically, we only consider the observed contexts in the training set. We calculate the Wasserstein distance between two random splits of the training set restricted to a subset of the context variables. Note that this linear program restricted to a subset of the context variables provides an upper bound to the solution of the linear program with full context, as the search space is smaller and the linear program is a minimization problem.

The dual variables convergence results for OPE and OPL problems using Wasserstein DRO are presented in Figure 3(a) and Figure 4(b) and the convergence results of using KL DRO are presented in Figure 3(b) and Figure 5(b). Additionally, the policy parameters learning curves of the OPL problems of two different methods are provided in 4(a) and 5(a), respectively.

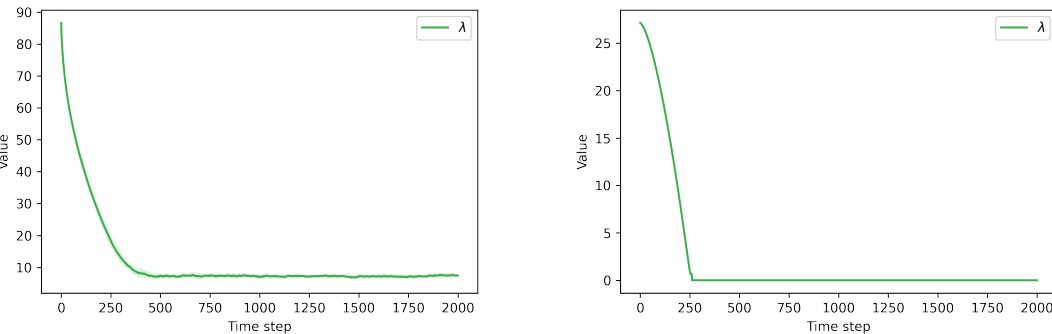

(a) Convergence curve of the dual variable of the regulated Wasserstein DRO (BSGD) ($\epsilon_{\mathrm{W}} = 0.03$).

(b) Convergence curve of the dual variable of the Factor KL DRO (gradient descent) ($\epsilon_{\mathrm{KL}} = 0.1$).

Figure 3: Policy evaluation convergence curves of the dual variables. The solid line and shades are averages and standard deviations over 20 runs.

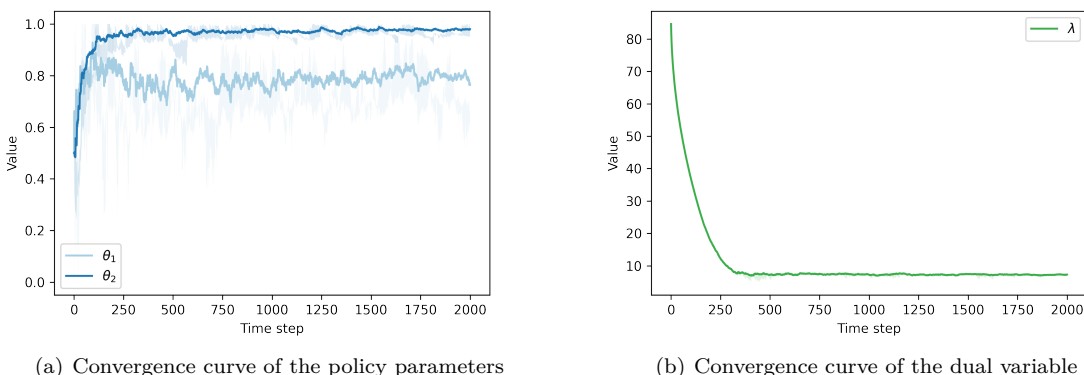

(a) Convergence curve of the policy parameters

(b) Convergence curve of the dual variable

Figure 4: Policy learning convergence results of the regulated Wasserstein DRO (BSGD) ($\epsilon_{\mathrm{W}} = 0.03$). The solid lines and shades are averages and standard deviations over 20 runs.

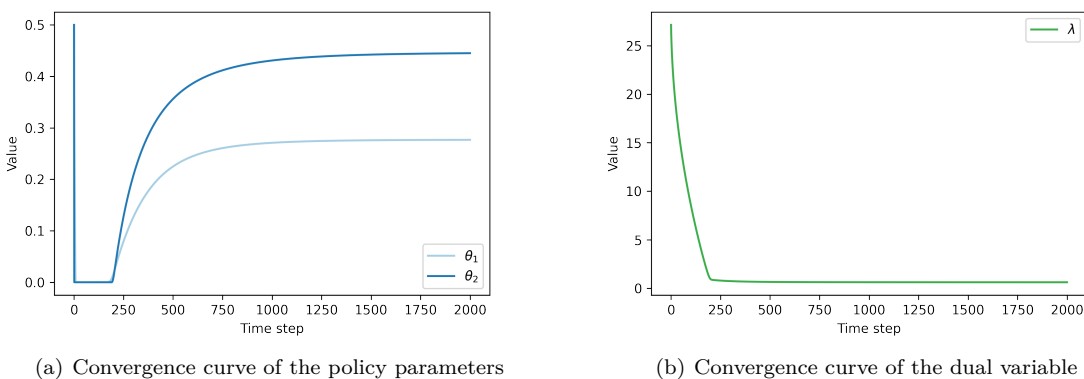

(a) Convergence curve of the policy parameters

(b) Convergence curve of the dual variable

Figure 5: Policy learning convergence results of the Factor KL DRO (gradient descent) ($\epsilon_{\mathrm{KL}} = 0.1$). There is no variance since the gradient descent method is deterministic.

