# OpenReview forum: "Wasserstein Distributionally Robust Policy Evaluation and Learning for Contextual Bandits"
_TMLR — Accepted by TMLR_

### Review · Reviewer_8pbY · 2023-10-27

**Summary Of Contributions:**

The paper proposes to use the Wasserstein distance instead of the KL in a Distributionally Robust formulation of Off-policy evaluation and off-policy learning for contextual bandits.
The paper is well organised: it starts from the direct definition of the problem  (2.1) and progressively highlights the computational issues and limitations that arise as we try to solve it. Through various clever ideas of regularizations and relaxation, an efficient estimator and algorithm is be provided with theoretical guarantees (Th 4.2 for the estimator, Th 5.1 for algorithm 1). Finally, an empirical study is done on medical data and showcases the benefits of the proposed method.

**Audience:**

Yes

**Broader Impact Concerns:**

None.

**Claims And Evidence:**

Yes

**Requested Changes:**

A pass on the text to clean up some typos and convoluted sentences.

Depending on the answer to my main question (major remarks), a modification may be needed in Algorithm 1.

**Strengths And Weaknesses:**

Strengths:
The paper is well-written despite a few typos, see minor remarks
Many contributions, including theoretical guarantees and empirical evaluation
The premise, that KL is not well suited for certain distribution shift tasks, is relevant and should be better known.

Major remarks:

In algorithm 1, is it not a problem that you use the same batch of sample to estimate the two gradients? I think you are introducing correlations and you should sample 2 separate batches. Can you please explain why you are allowed to do that?

Minor remarks:
Some typos are left so I recommend the authors to double check everything. I highlight a few below.
“ in practices”: incorrect

“ as shown in (Zhao & Guan, 2018) Proposition 2,”: awkward use of citation

“ The goal of the OPE problem is to bound the difference between Vˆ (π) and V”: no the goal of the OPE problem is to estimate the value of the policy offline. *Your* goal is to bound the difference between the estimator and V.

“Although the radius can be estimated when similar datasets that include both the testing and training sets are available, using the KL divergence, as discussed before, to calculate this radius poses additional challenges”: This sentence is upside down and must be rewritten

---

> ### Author Response · Authors · 2023-11-20
>
> Thank you for the constructive comments.  All new changes are marked in blue text in the revised manuscript.
>
> ---
>
> >Q1: In algorithm 1, is it not a problem that you use the same batch of samples to estimate the two gradients?
>
> Equations 5.3 and 5.4 calculate the gradient of $\hat{V}(\theta,\lambda)$ with respective to the variable $\theta$ and $\lambda$, respectively. It is using a single batch of samples to obtain the gradient of one function. In addition, in line 5 of Algorithm 1, the updating rule is not alternating between the policy variable $\theta$ and the dual variable $\lambda$. We believe this does not introduce correlations as the algorithm is using a single batch on a single function.
>
> ---
>
> > Requested changes
>
> Thanks for your suggestions and comments. We've rewritten all the sentences and fixed the typos in the revised manuscript.

---

### Review · Reviewer_Z8aE · 2023-11-05

**Summary Of Contributions:**

The paper studied off-policy evaluation (OPE) and off-policy learning (OPL) for contextual bandits under distribution shifts. The authors proposed  distributionally robust optimization (DRO) with Wasserstein distance, and its regularized version with a biased stochastic gradient descent algorithm for policy learning. The authors analyzed convergence rate and sample complexity guarantees. Empirical results on a stroke trial dataset validated the effectiveness of proposed Wasserstein DRO methods.

**Audience:**

Yes

**Broader Impact Concerns:**

No concerns.

**Claims And Evidence:**

Yes

**Requested Changes:**

Please see weaknesses and questions.

**Strengths And Weaknesses:**

Strengths

1. OPE and OPL under distribution shifts are important problems. The proposed DRO with Wasserstein distance is motivated to improve over previous works of DRO with KL divergence. The proposed Wasserstein DRO framework is intuitive.

2. Motivated by the high computation cost of Wasserstein DRO, the authors further proposed a regularized Wasserstein DRO problem and a biased stochastic gradient descent algorithm.

3. Besides the proposed new methods, theoretical analysis is another major contribution of the paper. Both convergence rate and sample complexity guarantees are analyzed for the proposed solutions.


Weaknesses and questions:

1. Assumptions. I appreciate that the authors clearly stated the assumptions. Assumption 3.3 is standard in OPE literature while assumption 3.1 is new to me. The necessity and benefit of this assumption are unclear: what would be the applications where assumption 3.1 is necessary? How does this assumption change the method/solution? Are new theoretical tools introduced to handle this assumption? I would also suggest the authors to further clarify the technical novelty compared to existing works in general.

2. Experiments. The authors are suggested to include details of KL DRO, and explain why not to compare with  KL DRO-based methods such as Si et al., 2020, Kallus et al., 2022, Mu et al., 2022. Regarding OPL, it is unclear why there is no baseline to compare with.

---

> ### Author Response · Authors · 2023-11-20
>
> Thank you for the constructive comments.  All new changes are marked in blue text in the revised manuscript.
>
> ---
>
> >Q1: On Assumption 2.2 (previous Assumption 3.3):
>
> This assumption is rooted in the dual formulation of the W-DRO problem. The problem will not be well-defined if we don't have this assumption. However, this assumption is not very strong, as you can set $y_{x,a}(\xi)$ to be $Y_{\max}$ if you don't know how $y$ relates $\xi$. Yet, this particular choice is quite conservative; we believe in many applications Assumption 2.2 holds and one can avoid setting the cost of all unobserved outcomes to be $Y_{\max}$. See the examples right after the Assumption 2.2 for details.
>
> Note that our theoretical results hold for any choice of $y_{x,a}(\xi)$. In practice, prior knowledge of $y_{x,a}(\xi)$ can help to reduce the conservativeness when applying DRO to OPE/OPL problems.
>
> In terms of novelty, the exact analysis is different from KL ones and cannot be quickly adapted from KL ones. We consider the geometry of the distributions, e.g., the support of the distribution, while KL does not. The proving strategy is similar, e.g., using concentrations, which is also the case for most of the finite sample analysis papers. In addition, we did not make extra assumptions on the dual variable, while [T. MU Factor DRO] further assumes that the dual variable is bounded from below. More importantly, we provide a practical algorithm to numerically solve the optimization problem and show the iteration and query complexity of the proposed algorithm. The algorithmic part is mostly missing in previous KL papers.
>
> ---
>
> > Q2: Experiments
>
> Thanks for the suggestions. The KL DRO method in the experiments is adopted from [Mu Factor DRO 2022]; we renamed the KL method and put more details on it; see the blue texts in Section 6 in the revised manuscript. We did not compare with the KL DRO methods in [Si et al., 2020] and [Kallus et al., 2022] as they consider the distribution shifts from contexts and rewards as a whole, while we separate the two sources of shifts (the same as in [Mu Factor DRO 2022]). We believe the KL DRO method proposed in [Mu Factor DRO 2022] is a more meaningful baseline method.
>
> We added the baseline result of the Factor KL DRO on OPL problems. The results can be found in Table 1 and more experimental details are further provided in Section F of the Appendix in the revised manuscript.

---

> > ### Comment · Reviewer_Z8aE · 2023-12-04
> >
> > I thank the authors for the response. My concerns are addressed.

---

### Review · Reviewer_oaWR · 2023-11-05

**Summary Of Contributions:**

This study considers distributionally robust optimization (DRO) in off-policy evaluation (OPE) and off-policy learning (OPL). First, in Definition 2.1, the authors define the distributionally robust policy value $V(\pi)$ as
$$V(\pi) := \sup_{P_x}E_{x\sim P_x}[E_{a\sim\pi(\cdot|x)}[\sup_{P_{x,a}}E_{P_{x,a}}[Y]] ],$$
where the supremums are taken over perturbations within small deviations computed by the Wasserstein distance.
Then, they propose OPE and OPL algorithms based on the policy value with strong theoretical guarantees. Theorem 3.4 and 5.1 show seemingly tight upper bounds for OPE and OPL, respectively.

**Audience:**

Yes

**Broader Impact Concerns:**

None.

**Claims And Evidence:**

Yes

**Requested Changes:**

I felt that this manuscript is a bit hard to read because the problem setting is unclear in the sense of OPE and OPL. In OPE and OPL, the difference between the potential outcomes and observations plays an important role. In this sense, I believe that observations should be described in the problem-setting section (Section 2). However, they are described in Assumption 3.1 with a new variable $\xi$. It would be more desirable if these definitions were mentioned in the problem-setting section.

Furthermore, the overall assumptions are incrementally added as the section goes on, which makes this study hard to understand. Some assumptions are more related to the problem setting itself, as well as the difference between the potential outcomes and observations. For example, the knowledge about the functional form of $y(\cdot)$ and the existence of $xi$ should be clarified in Section 2.

Additionally, the authors should clarify which variables are known or unknown before the OPE and OPL. For example, is $\pi$ known to us before the OPE and OPL?

Minor comments.
1. page 5: set(Hu & Hong, 2013) -> set (Hu & Hong, 2013) (space).

**Strengths And Weaknesses:**

The strength of this study lies in the proposal of the distributionally robust policy value and algorithms appropriately designed for the policy value. The authors effectively highlight the issue of distributional overlap in DRO and introduce a new policy value that addresses this concern. Subsequently, they propose OPE and OPL algorithms and reveal the theoretical properties. The paper makes insightful contributions, and I am inclined to support its acceptance.

Despite the valuable contributions, I have two principal concerns:

1. On the necessity of considering DRO in OPE and OPL.

I am curious when the problem setting is useful. For example, in OPE and OPL, we usually assume that $E[Y|X]$ is identifiable. Once $E[Y|X]$ is identified, it is at least robust for distributional change in $X$. Furthermore, if we do not know $E[Y|X]$ and do not assume common support, it seems that OPE and OPL become infeasible. I mean that the necessity of DRO in OPE and OPL significantly depends on the underlying assumptions. I would like the authors to clarify these points, especially what assumptions are required.

2. On the use of the Wasserstein distance.

As the authors discuss, the use of the Wasserstein distance is necessary when there is no overlap support, however, without any assumptions on either the functional form of the costs. In fact, the authors assume some restrictions or knowledge on the functional form. I believe that there is a tradeoff between weakening the overlap assumptions and imposing additional assumptions on the related random variables.

While the definition of policy value and the methods proposed are persuasive, the study's framework seems somewhat contrived within the context of OPE and OPL. It appears that the problem setting was adapted to incorporate the Wasserstein distance within OPE and OPL rather than applying the Wasserstein distance to address issues within the existing problem setting. The presentation would be improved by clarifying what assumptions are the same and what assumptions are different compared with the original OPE and OPL problems.

---

> ### Author Response · Authors · 2023-11-20
>
> Thank you for the constructive comments.  All new changes are marked in blue text in the revised manuscript.
>
> ---
> > Q1. On the necessity of considering DRO in OPE and OPL.
>
> DRO should be applied to OPE/OPL problems whenever there are distribution shifts between the training set and the testing set, where the training set is recorded as an offline dataset and the testing set is unknown. If you can observe the testing set, then DRO is not necessary as conventional OPE/OPL methods can be used directly.
>
> DRO OPE/OPL problems share some common assumptions with standard OPE/OPL problems, see Assumption 2.1 (previous 3.1) in the revised manuscript.
>
> When applying KL-DRO to OPE/OPL problems, an extra assumption on the probability mass function of the cost functions is required. In particular, in KL-DRO papers, it requires that the probability of $P(Y=y|x,a)$ is lower bounded for all $y$ (assume $y$ is discrete) and across all context and action pairs. This lower bound pmf assumption is related to the dual problem of the KL DRO. (We added some comments on this in Section 3 after the Proposition 3.1.)
>
> Under the Wasserstein-DRO (W-DRO) formulation, we also need (similar) extra assumptions to ensure that the dual problem under W-DRO is well-defined, which is Assumption 2.2 (previous 3.3) in the revised manuscript.
>
> ---
> > Q2. On the use of the Wasserstein distance.
>
> i) For a fix context and action pair (x,a), we assume that a function $y_{x,a}(\xi)$ is given and $\xi$ is a random variable with unknown distribution. Your question is about whether the function form of $y_{x,a}$ can help us infer the values of unobserved $\xi$ (non-overlap points).
>
> Currently, we only consider the discrete setting (finite many $\xi$), the function form does not allow us to apply some knowledge from observed data points to unobserved data points since $y_{x,a}(\xi)$ can be defined arbitrarily. In this sense, the functional form does not "relate" random variables.
>
> We think it will be an interesting topic to study when the support is continuous and the sub-optimality gap might depend on the assumption of the functional form. However, the continuous case is not a straightforward extension. To see this, discrete contexts and actions allow us to compute the robust policy value for each context and action separately pair as shown in equation 2.1. Any regularity conditions, e.g., smoothness or lipschitzness, will make the problem much harder as the context action pairs are coupled. We leave this as future work.
>
> ii) We believe the biggest difference is that the previous OPE and OPL problems using KL divergence cannot include out-of-support distributions in the uncertainty set. For example, if the testing distribution contains new contexts, KL-DRO fails to include the testing distribution in the uncertainty set regardless of the size of the radius. As a result, Wasserstein distance addresses the out-of-support issue that cannot be solved by previous works.
>
> ---
> > Requested changes
>
> Thanks for the suggestions, we've moved the assumptions to Section 2.
>
> We also clarified the information on $\pi$ in the theorems related to OPE and OPL problems.

---

### Decision · Action_Editor_ogSJ · 2023-12-17

**Recommendation:** Accept as is

**Comment:**

I agree with the reviewers that the paper is providing a very well-rounded contribution to the problems of policy evaluation/optimization in contextual bandits under distribution shifts. Especially, this work seems to properly address some prior works limitations, which either model the distribution shift through KL divergence or do not provide comparable theoretical results for Wasserstein DRO.

While one reviewer is concerned by the relevance of the problem setting, I personally think it is sufficiently motivated and that the contribution will interest a sub-community of TMLR.
Thus, I am recommending to accept the paper as is.

I am further proposing a "featured certification" in view of its technical quality, which has been mostly acknowledged by all the reviewers.

**Audience:**

This work is at the intersection of distributionally robust optimization, RL theory, and bandits. I think it can draw interest of a meaningful TMLR community.

**Claims And Evidence:**

The claim of the papers are well supported by formal proofs of the theoretical results and an empirical comparison between Wasserstein and KL DRO.

---

> ### Author Response · Authors · 2024-01-15
> **Thank you all**
>
> We thank the editors and the reviewers for the constructive comments and suggestions!